# FEDERATED ACTIVE LEARNING VIA CLASS-ADAPTIVE LOCAL–GLOBAL BALANCING

## ABSTRACT

Active learning has emerged as a pivotal approach for addressing data scarcity and annotation cost constraints in machine learning systems. However, its implementation in federated learning settings introduces unique challenges, particularly concerning data heterogeneity across clients. Our comprehensive analysis of existing centralized and decentralized methodologies reveals that state-of-the-art federated active learning techniques do not always outperform simpler baselines where centralized techniques are applied independently to clients. We identify a critical trade-off in performance: decentralized approaches excel when inter-client data heterogeneity is minimal, while centralized methods demonstrate superior performance under high heterogeneity conditions. Moreover, we observe a class-dependent variance phenomenon where the efficacy of each approach strongly correlates with the distribution variance of class samples across federated clients, highlighting critical bounds that limit existing methods. To address these limitations, we propose Adaptive Hybrid Federated Active Learning (AHFAL), a novel framework that dynamically integrates centralized and decentralized paradigms based on class-specific distribution characteristics. AHFAL combines enhanced entropy-based sampling with heterogeneity mitigation strategies, adaptively selecting the optimal paradigm per class based on cross-client variance metrics. Experiments across diverse datasets demonstrate that AHFAL outperforms state-of-the-art methods by prioritizing heterogeneity management over traditional uncertainty sampling, particularly in low-resource and high heterogeneity scenarios.

## 1 INTRODUCTION

Federated learning (FL) has emerged as a compelling paradigm for collaborative model training across distributed clients (McMahan et al.; Konečný et al., 2016). However, FL commonly assumes access to sufficiently large labeled datasets at each client, which is often unrealistic due to annotation costs and required expert knowledge (Litjens et al., 2017). Active learning (AL) addresses data scarcity by iteratively selecting the most informative samples for annotation (Settles, 2009; Ren et al., 2021). Federated active learning (FAL) combines FL and AL to enable collaborative, data-efficient, and privacy-preserving learning when labeled data are scarce and centralized data pooling is infeasible (Cao et al., 2023; Kim et al., 2023; Chen et al., 2024).

Classical AL methods (e.g., BADGE (Ash et al., 2019), Entropy (Holub et al., 2008), and Core-Set (Sener & Savarese, 2018)) assume access to the complete dataset and use metrics such as representativeness or uncertainty as proxies for informativeness. In federated settings, these assumptions do not hold: client datasets are partitioned in a non-i.i.d. manner, labeling budgets are allocated per client, and no party has global visibility of all samples. These conditions make sample selection considerably harder in FAL than in classical settings.

We systematically investigate centralized methods (where clients apply traditional AL methods independently) and decentralized methods, which leverage cross-client information. Our analysis uncovers three critical insights into how sample selection operates in FAL. First, aggregate heterogeneity determines which methods prevail: decentralized approaches excel when client distributions are similar, centralized approaches dominate under strong heterogeneity. Second, the crossover is explained at the class level: high-variance classes concentrated on a few clients benefit from centralized querying, and low-variance classes with broad coverage gain from inter-client information

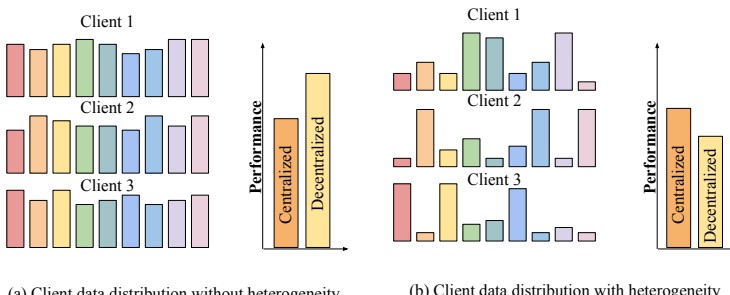

(a) Client data distribution without heterogeneity      (b) Client data distribution with heterogeneity

**Figure 1:** Prior work in active learning divides into centralized methods (operating independently per client) and decentralized methods (utilizing both local and global information). Our analysis reveals a crucial trade-off: (a) decentralized methods excel when cross-client data heterogeneity is low, while (b) centralized methods surprisingly outperform when heterogeneity is high—even surpassing methods specifically designed for federated settings. Our approach leverages this insight by treating data heterogeneity as the key performance determinant, enabling robust results especially for high heterogeneity levels through adaptive sampling.

sharing. Third, aligning local sampling with the global class distribution consistently improves accuracy, showing that mitigating heterogeneity can be more impactful than refining heuristics.

To operationalize these findings, we propose Adaptive Hybrid Federated Active Learning (AHFAL), a class-adaptive framework that dynamically toggles between centralized and decentralized sampling methods on a per class basis. AHFAL estimates global class distribution, quantifies per-class variance across clients, and assigns classes to either low- or high-variance regimes. For low-variance classes, it aggregates entropy estimates from local and global models; for high-variance classes, it prioritizes local model predictions. Sample selection is further refined through class-aware budget allocation, prioritizing rare and underrepresented classes. Our key contributions are threefold:

1. We provide a systematic analysis of centralized and decentralized FAL methods, uncovering three critical insights: (i) aggregate heterogeneity determines whether centralized or decentralized methods are more effective, (ii) class-wise variance explains the performance crossover, and (iii) global distribution knowledge outweighs fine-grained informativeness heuristics.

2. Building on these insights, we present Adaptive Hybrid Federated Active Learning (AHFAL), a novel algorithm that adaptively selects sampling strategies based on class-wise variance.

3. We demonstrate through extensive experiments that AHFAL consistently outperforms prior FAL methods, with the strongest gains in high-heterogeneity regimes.

These findings establish client heterogeneity, especially class-wise variance, as the primary challenge in FAL, motivating adaptive methods that tailor sampling strategies to heterogeneity conditions.

## 2 RELATED WORK

### 2.1 ACTIVE LEARNING

Most data available for machine learning is unlabeled, and acquiring labels is costly, time-consuming, and often requires domain expertise. AL addresses this challenge by selecting the most informative samples for annotation (Settles, 2009; Schröder & Niekler, 2020). AL strategies can be broadly divided into two categories: First, uncertainty-based methods (Scheffer et al., 2001; Gissin & Shalev-Shwartz, 2019; Lewis, 1995; Ranganathan et al., 2017; Sinha et al., 2019; Ducoffe & Precioso, 2018; Mayer & Timofte, 2020) select samples where the model exhibits high predictive uncertainty, typically near decision boundaries. Second representation- and diversity-based methods (Wu et al., 2006; Ienco et al., 2013; Kang et al., 2004; Elhamifar et al., 2013; Hu et al., 2010; Sener & Savarese, 2017; Shui et al., 2020) exploit the structure of the unlabeled data to select samples that best capture the structure of the input space. However, recent work demonstrates that no single AL method is universally optimal: performance depends on dataset characteristics, task complexity, and labeling budgets. This has motivated adaptive AL methods, which dynamically select among strategies during training (Hacohen & Weinshall, 2023; Zhang et al., 2023; Hsu & Lin, 2015; Pang et al., 2018).

## 2.2 FEDERATED ACTIVE LEARNING

FAL extends the core principles of FL (Hsu et al., 2019; Konečný et al., 2016; McMahan et al., 2017; Chen & Chao, 2021; Hsu et al., 2020; Mohri et al., 2019; Gong et al., 2021; Lin et al., 2020) by enabling clients to query samples for annotation while models are trained collaboratively. In FAL, decentralized methods combine local and global information to guide selection. LoGo (Kim et al., 2023) introduced a two-stage, cluster-wise selection combining gradient embeddings from a local model with uncertainty scoring from a global model to balance intra-client diversity and global minority classes. FEAL (Chen et al., 2024) models aleatoric and epistemic uncertainties with a Dirichlet evidential head. LeaDQ (Sun et al., 2025) frames active querying as a decentralized POMDP to learn per-client policies. KAFAL (Cao et al., 2023) tackled sampling aggregation mismatches by reweighting class-specific discrepancies to mitigate aggregation mismatches. Despite these advances, existing decentralized methods remain constrained by predefined heuristics and fixed global–local fusion rules. While adaptive methods have proven effective in centralized AL, extending this perspective to federated settings (where data heterogeneity and communication constraints pose additional challenges) remains largely unexplored. Our work addresses this gap by proposing an adaptive framework based on data conditions.

## 3 PROBLEM FORMULATION

We consider a federated system with $N$ clients. Client $i$ has a labeled set $\mathcal{L}_i = \{(x_j, y_j)\}_{j=1}^{|\mathcal{L}_i|}$ and an unlabeled pool $\mathcal{U}_i = \{x_j\}_{j=1}^{|\mathcal{U}_i|}$, where $x_j \in \mathcal{X}$ and $y_j \in \mathcal{Y} = \{1, \dots, C\}$. Each client trains a local model $f_{\theta_i}^{\mathrm{L}}$, and the server maintains a global model $f_\theta^{\mathrm{G}}$ via aggregation.

At each active learning round, a budget of $B$ queries is available across the federation. The learner selects

$$\mathcal{S} = \bigcup_{i=1}^{N} \mathcal{S}_i, \quad \mathcal{S}_i \subseteq \mathcal{U}_i, \quad |\mathcal{S}| = B,$$

whose labels are revealed and added to the local sets. The optimal selection minimizes test error:

$$\mathcal{S}^\star = \arg\min_{\mathcal{S}} \; \mathbb{E}_{(x,y)\sim\mathcal{P}_{\mathrm{test}}}\big[\mathcal{L}(f_{\theta(\mathcal{S})}(x), y)\big], \tag{1}$$

where $\theta(\mathcal{S})$ are the parameters obtained after federated training on $\bigcup_i (\mathcal{L}_i \cup \mathcal{S}_i)$, and $\mathcal{L}(\cdot, \cdot)$ denotes the task loss; in our experiments we evaluate using accuracy.

Since raw data remain local, $\mathcal{S}$ must be chosen from local features, predictions, and aggregate statistics broadcast by the server. We next analyze how these constraints interact with client heterogeneity.

## 4 EMPIRICAL ANALYSIS: ACTIVE LEARNING UNDER CLIENT HETEROGENEITY

We present illustrative experiments to highlight how client heterogeneity affects FAL. These findings motivate our mathematical analysis and the design of AHFAL.

### 4.1 EXPERIMENTAL SETUP

We conduct experiments on CIFAR-10 Krizhevsky et al. (2009). Clients are partitioned using the Dirichlet scheme with concentration parameters $\alpha \in \{0.05, 0.1, 0.3, 0.5, 1, 10\}$ ranging from highly skewed to near-IID regimes. A ResNet-8 backbone is trained locally, with updates aggregated via FedAvg. At each round, clients acquire 5% of labels using the given sampling strategy. Performance is measured by test accuracy as a function of the labeled-data budget. We compare against two categories of sampling strategies:

- **Centralized baselines** (run *locally* on each client): ENTROPY (Holub et al., 2008), BADGE (Ash et al., 2019), CORE-SET (Sener & Savarese, 2018), and NOISE STABILITY (Li et al., 2024).

- **Decentralized baselines** (*global-aware*): LOGO (Kim et al., 2023), FEAL (Chen et al., 2024) and KAFAL (Cao et al., 2023).

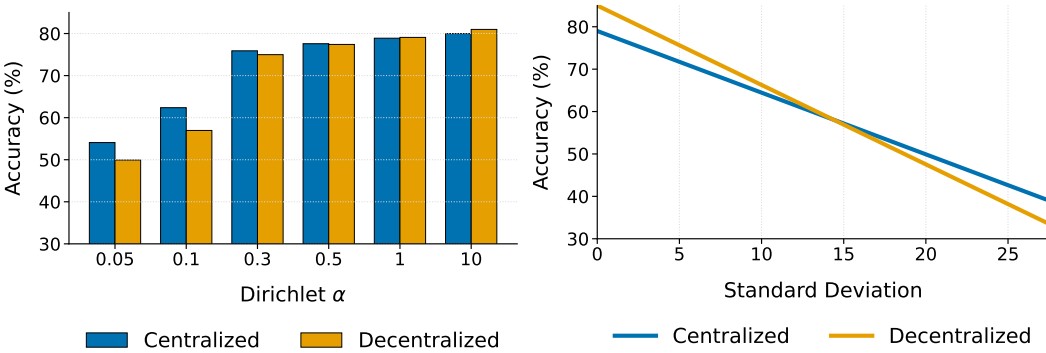

**Figure 2: Aggregate heterogeneity tradeoff.** Decentralized strategies excel when client distributions are similar (large $\alpha$), while centralized methods dominate under strong heterogeneity (small $\alpha$).

**Figure 3: Class-wise variance explains the crossover.** Classes with high $CV_c$ favor centralized sampling, while low-variance classes benefit from decentralized selection. Each line is a least-squares fit.

### 4.2 KEY FINDINGS

Our analysis yields three key findings on the role of heterogeneity in FAL:

**Finding 1: Aggregate heterogeneity drives the centralized–decentralized trade-off (Figure 2).**

We find that the relative effectiveness of centralized and decentralized method is not universal but regime-dependent. Decentralized strategies outperform when client data is similar (large $\alpha$), whereas centralized strategies relying only on local data dominate when heterogeneity is high (small $\alpha$). No static strategy is effective across all regimes.

**Finding 2: Class-wise variance explains the crossover (Figure 3).**

We uncover that the performance crossover is driven at the class level. To quantify how unevenly a class $c$ is distributed, we compute its coefficient of variation $CV_c = \frac{\sigma_c}{\mu_c}$, where $\{n_{i,c}\}_{i=1}^{N}$ are the client-wise counts of class $c$, $\mu_c$ is their mean, and $\sigma_c$ their standard deviation. High-variance classes (large $CV_c$), concentrated on a few clients, benefit from centralized querying, whereas low-variance classes (small $CV_c$), broadly distributed across clients, perform best using decentralized methods. This reveals class-wise variance as the mechanism underlying the aggregate crossover. Note that for Figure 3, we compute inter-client standard deviation, test accuracy across all classes, $\alpha$ values, methods, and seeds, and plot only the resulting linear trends.

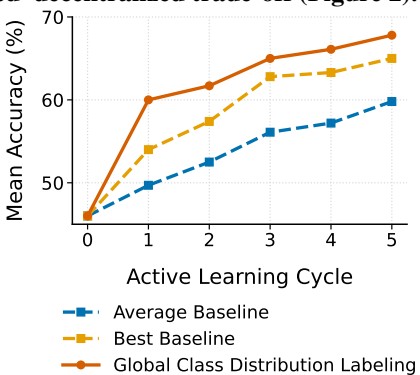

**Figure 4: Oracle experiment.** Providing each client with the target class histogram (no raw data) yields a consistent 2–3% accuracy lift, showing that *heterogeneity reduction*, not finer heuristics, is the dominant lever.

**Finding 3: Global distribution knowledge outweighs finer uncertainty estimates (Figure 4).**

Finally, we test an oracle scenario where each client is provided with the true global class distribution (but no raw data). Clients adjust their queries to narrow the divergence between their local and global histograms. As shown in Figure 4, this simple alignment yields a consistent 2–3% accuracy gain across sampling heuristics (e.g., entropy, typicality). This confirms that mitigating heterogeneity is more impactful than refining uncertainty estimates.

**Takeaway.** *Client heterogeneity, especially at the class level, is the principal obstacle in federated active learning.* A practical method must (i) detect client distribution heterogeneity (with regards to the global distribution) as well as class-wise variance and (ii) adapt its sampling policy accordingly: precisely the design principles embodied by **AHFAL**.

## 5 THEORETICAL INSIGHTS

To explain the empirical findings in Section 4, we study entropy estimation under client heterogeneity. Our goal is to relate classwise performance to inter-client variance for each class $c$, comparing decentralized (global-aware) and centralized (local-only) scoring.

**Two forces that determine error.** We model acquisition scoring as estimating the Bayes predictive entropy and analyze how client heterogeneity affects estimator error (details in Appendix C). Two effects govern performance for a class $c$ on client $i$: (i) the variance of the local estimator, which decreases with the client's class count $n_{i,c}$, and (ii) the global estimator's class bias $\beta_c$, which grows with cross-client imbalance (captured by the dispersion $\sigma_c$: the cross-client standard deviation of the class-$c$ proportions.).

**Why and when to average local and global entropies as a measure of uncertainty.** We consider a convex combination of local and global entropies. The optimal weight minimizes the MSE of the ensemble and reduces to a simple classwise decision between *local* ($\lambda=1$) and a fixed *hybrid* ($\lambda=1/2$) estimator. Writing $V_L, V_G$ for the per-class variances and $\rho$ for their covariance (all w.r.t. $x \sim \mathcal{D}_{i,c}$), hybrid improves over local whenever

$$\beta_c^2 \;<\; 3\,V_L \;-\; V_G \;-\; 2\,\rho,$$

and local is otherwise preferred (see Appendix C for the derivation). Practically, $\beta_c$ is unobserved; we use $\sigma_c$ as a proxy (monotonicity assumption).

Moreover, the full MSE expression for the convex combination $\hat{H}_c^{(\lambda)}(x) = \lambda \hat{H}_c^L(x) + (1-\lambda)\hat{H}_c^G(x)$ (derived in Appendix C) admits a closed-form minimizer $\lambda^\star \in (0,1)$ whenever the global bias $\beta_c$ is not too large. In the *symmetric* regime where local and global estimators have comparable bias and variance (precisely the low-variance classes where condition above holds), $\lambda^\star$ concentrates near $1/2$, and the MSE is quadratic in $\lambda$, so the excess error scales as $(\lambda - \lambda^\star)^2$. Thus the fixed choice $\lambda = 1/2$ used in Eq. equation 3 is a simple, closed-form surrogate that is near-optimal throughout the regime where hybridization is preferable, while avoiding the need to estimate class- and client-specific mixing weights.

**Takeaway.** For *high-heterogeneity* classes (large $\sigma_c$) on *data-rich* clients, local scoring dominates; for *low-heterogeneity* classes or *client-poor* situations, the hybrid estimator reduces error. This aligns with—and explains—the empirical crossovers reported in Section 4.

**Connection to AHFAL.** The MSE analysis above shows that for each class $c$ there is a threshold condition

$$\beta_c^2 < 3V_L - V_G - 2\rho$$

under which a hybrid entropy estimator has lower error than a purely local one, and the opposite regime where local entropy is preferable. Since the global class bias $\beta_c$ is not directly observable, we use the empirical cross-client class variance $\sigma_c$ as a monotone proxy for $\beta_c^2$. AHFAL implements a discretized version of this criterion by partitioning classes into $\mathcal{C}_{\text{low}} = \{c : \sigma_c < \tau\}$ and $\mathcal{C}_{\text{high}} = \{c : \sigma_c \geq \tau\}$: classes in $\mathcal{C}_{\text{low}}$ use a fixed hybrid entropy $(H^L + H^G)/2$, while classes in $\mathcal{C}_{\text{high}}$ use purely local entropy $H^L$. Thus the local-versus-hybrid routing rule and fixed mixing weight in Eq. equation 3 are a direct operationalization of the MSE-based condition above.

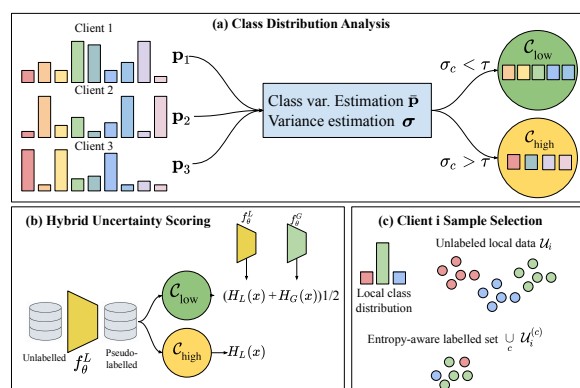

**Figure 5:** AHFAL consists of 3 steps: (a) the global class distribution, class variances and class partitioning into low and high variance groups is calculated and broadcasted by server; (b) the hybrid uncertainty scoring is carried out as a function of class variance; (c) class-aware sample allocation is carried out based on uncertainty scores for unlabeled samples.

# 6 ADAPTIVE HYBRID FEDERATED ACTIVE LEARNING (AHFAL)

We now present **AHFAL**, a class-adaptive framework for federated active learning that integrates centralized and decentralized sample selection by leveraging class-specific distributional statistics. Motivated by the observed correlation between per-class distribution variance and optimal selection strategy, AHFAL explicitly quantifies heterogeneity at the class level and adjusts its sampling paradigm accordingly. Figure 5 shows the overall AHFAL method.

## 6.1 AHFAL SAMPLE SELECTION

**Step 1: Class Distribution Analysis**

Motivated by Finding 1, AHFAL estimates global class statistics to capture per-class variance. Let $\mathcal{L}_i \subset \mathcal{D}_i$ denote the labeled dataset at client $i$, initially comprising 10% of $\mathcal{D}_i$, obtained via random sampling. Each client computes its empirical class distribution vector $\mathbf{p}_i = \left[\frac{n_{i,1}}{|\mathcal{L}_i|}, \dots, \frac{n_{i,C}}{|\mathcal{L}_i|}\right]$, where $n_{i,c}$ is the number of labeled examples of class $c$ in $\mathcal{L}_i$ and $C$ is the number of classes. Clients transmit $\mathbf{p}_i$ to the central server, which computes the mean class distribution $\bar{\mathbf{p}}$, defined as $\bar{p}_c = \frac{1}{N}\sum_{i=1}^{N} p_{i,c}$, and the standard deviation vector $\boldsymbol{\sigma}$, defined as $\sigma_c = \sqrt{\frac{1}{N}\sum_{i=1}^{N}(p_{i,c} - \bar{p}_c)^2}$ for class $c$. These serve as the target distribution and class variance estimators, respectively.

Classes are partitioned into two disjoint sets:

$$\mathcal{C}_{\text{low}} = \{c \in \{1, \dots, C\} \mid \sigma_c < \tau\}, \qquad \mathcal{C}_{\text{high}} = \{1, \dots, C\} \setminus \mathcal{C}_{\text{low}} \tag{2}$$

where $\tau$ is a fixed variance threshold. This partitioning dictates whether sample selection for class $c$ should be informed by global model predictions ($c \in \mathcal{C}_{\text{low}}$) or rely solely on the local model ($c \in \mathcal{C}_{\text{high}}$).

**Step 2: Hybrid Uncertainty Scoring** From Finding 2, AHFAL adapts uncertainty scoring based on class-wise variance. Each client forwards its unlabeled pool $\mathcal{U}_i$ through its local model $f_{\theta_i}^L$ to generate pseudo-labels and compute predictive entropy $H^L(x) = -\sum_{c=1}^{C} f_{\theta_i}^L(x)_c \log f_{\theta_i}^L(x)_c$. For classes $c \in \mathcal{C}_{\text{low}}$, clients also query the global model $f_\theta^G$ to obtain entropy $H^G(x)$. The final uncertainty score is defined as:

$$H(x) = \begin{cases} H^L(x), & \text{if } \hat{y}(x) \in \mathcal{C}_{\text{high}} \\ \frac{1}{2}(H^L(x) + H^G(x)), & \text{if } \hat{y}(x) \in \mathcal{C}_{\text{low}} \end{cases} \tag{3}$$

where $\hat{y}(x) = \arg\max_c f_{\theta_i}^L(x)_c$ denotes the pseudo-label.

The fixed mixing weight $\frac{1}{2}$ is chosen following the MSE analysis in Section 5, which shows that for low-variance classes (where $\hat{H}_c^L$ and $\hat{H}_c^G$ have comparable bias and variance) the MSE-optimal weight $\lambda^\star$ lies near $1/2$, and the MSE penalty for using this symmetric value is second-order in $(\lambda - \lambda^\star)$.

While we instantiate $H^L(x)$ and $H^G(x)$ using predictive entropy in our experiments, AHFAL is agnostic to the specific acquisition function: any scalar uncertainty score (e.g., margin sampling, BALD, or variation ratios) can be used in place of entropy without changing the class-variance-based routing or budget allocation steps. We leave a more detaied analysis of this to future work.

**Step 3: Class-Aware Budget Allocation and Sample Selection**

Motivated by Finding 3, AHFAL allocates budgets to align queries with the global distribution. Let $B_i$ denote the client's sample selection budget. To reduce local-global divergence, each client computes a target count vector $\mathbf{b} = [b_1, \dots, b_C]$ for selecting samples by minimizing the discrepancy between the local and global class distributions. The class-wise budget is determined by:

$$b_c \propto \begin{cases} 1, & \text{if } n_{i,c}^{\text{labeled}} = 0 \\ \frac{1}{n_{i,c}^{\text{labeled}}}, & \text{otherwise} \end{cases} \tag{4}$$

subject to the constraint $\sum_{c=1}^{C} b_c = B_i$. This encourages selecting underrepresented/missing classes.

For each class $c$, the client identifies the subset $\mathcal{U}_i^{(c)} \subset \mathcal{U}_i$ of pseudo-labeled samples with $\hat{y}(x) = c$, ranks them by entropy $H(x)$ in descending order, and selects the top $b_c$ samples. If $\mathcal{U}_i^{(c)}$ contains fewer than $b_c$ eligible samples, the deficit is redistributed proportionally to underrepresented classes.

## 6.2 TYING INTO THE FEDERATED LEARNING PIPELINE

We now describe how AHFAL fits into the broader FL pipeline. In practice, these selection steps are interleaved with the standard federated optimization loop. Concretely, the system proceeds in **rounds**. Each round comprises:

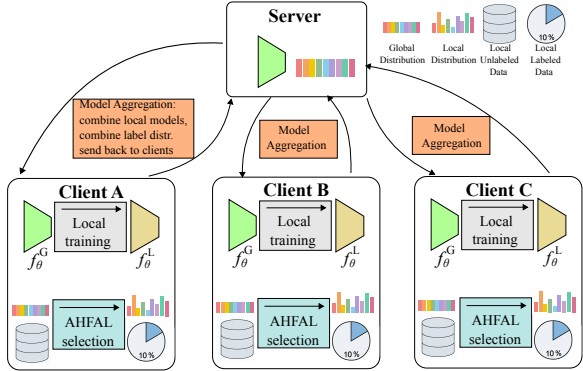

**Figure 6:** One federated learning round consists of local training, AHFAL selection, and model aggregation.

1. **Local training**: every client performs $E$ epochs of training on its current labeled set $\mathcal{L}_i$ and ships the updated weights to the server;

2. **Model aggregation**: the server aggregates the weights to yield the new global model $f_\theta^G$;

3. **AHFAL selection**: clients compute class statistics, partition classes into $\mathcal{C}_{\text{low}}/\mathcal{C}_{\text{high}}$, score their unlabeled pools with $H(\cdot)$, and acquire $B$ additional labels.

An additional computational cost arises from forward passes over the unlabeled pool $U_i$ on each client to compute uncertainty scores, which scales linearly with the pool size, i.e., $O(|U_i|)$. This overhead is lightweight compared to local training and requires no extra communication. No raw data is exchanged at any point; only model updates and aggregated class statistics are shared. Figure 6 illustrates the integration of AHFAL into the federated learning loop.

## 6.3 PRIVACY CONSIDERATIONS

Sharing class distributions with the server may introduce potential privacy risks.

To mitigate these risks, we consider two complementary mechanisms. First, we adopt *local differential privacy*, where each client perturbs its class histogram with calibrated Laplace noise before communication (Setlur et al., 2025; Suresh, 2019). The overall privacy budget $\varepsilon$

**Table 1:** Test accuracy (%) comparison across methods and data heterogeneity on CIFAR-10.

| Method | $\alpha = 0.1$ | $\alpha = 0.3$ | $\alpha = 0.5$ | $\alpha = 1.0$ |
|---|---|---|---|---|
| Random | $56.25 \pm 3.73$ | $74.00 \pm 1.58$ | $76.72 \pm 0.62$ | $77.71 \pm 0.44$ |
| Entropy | $64.23 \pm 3.48$ | $76.89 \pm 1.22$ | $78.99 \pm 0.60$ | $80.16 \pm 0.47$ |
| BADGE | $61.01 \pm 1.37$ | $75.00 \pm 0.90$ | $76.32 \pm 0.77$ | $77.87 \pm 0.26$ |
| Core-Set | $64.21 \pm 1.20$ | $76.40 \pm 0.61$ | $77.35 \pm 0.22$ | $79.00 \pm 0.41$ |
| Noise Stability | $60.04 \pm 3.93$ | $75.26 \pm 1.12$ | $77.64 \pm 0.60$ | $78.54 \pm 0.17$ |
| LoGo | $58.22 \pm 4.98$ | $74.95 \pm 1.62$ | $77.18 \pm 0.45$ | $79.06 \pm 0.72$ |
| KAFAL | $55.57 \pm 4.75$ | $74.16 \pm 1.06$ | $77.16 \pm 0.91$ | $79.25 \pm 0.72$ |
| FEAL | $57.08 \pm 1.98$ | $75.83 \pm 1.81$ | $77.88 \pm 0.22$ | $78.93 \pm 0.40$ |
| AHFAL (Ours) | $\mathbf{66.15 \pm 0.94}$ | $\mathbf{77.26 \pm 0.45}$ | $\mathbf{79.10 \pm 0.47}$ | $\mathbf{79.82 \pm 0.39}$ |

can be distributed across active learning cycles, ensuring rigorous privacy guarantees. Since noise is applied to class histograms rather than to raw data or model gradients, its effect on accuracy is only indirect.

This contrasts with differential privacy applied directly to data or gradients, which typically has a stronger impact on utility. As a result, the privacy–utility trade-off in our setting is considerably more favorable. Second, we consider *secure aggregation* of class histograms, in which clients encrypt their local class statistics such that the server only observes the aggregate sum, never any individual contributions. This prevents reconstruction of single-client distributions while preserving full utility. Prior work has shown secure aggregation to be highly efficient even for high-dimensional vectors (Bonawitz et al., 2017); in our case, the exchanged histograms are low-dimensional, making the overhead minimal. Together, these mechanisms provide complementary options: local DP offers provable privacy at the cost of controlled noise, while secure aggregation eliminates per-client leakage without affecting accuracy. We defer the empirical evaluation of local differential privacy to Section 7.3.

## 7 RESULTS

We now evaluate AHFAL, and compare it against baseline methods across datasets, client heterogeneity, model architecture as well as privacy budgets.

### 7.1 EXPERIMENTAL SETUP

**Datasets and Partitioning.** We evaluate on CIFAR-10 (Krizhevsky et al., 2009), CIFAR-100 (Krizhevsky et al., 2009), SVHN (Netzer et al., 2011), and MNIST (Le-Cun et al., 2010). Client data is partitioned using the standard Dirichlet scheme (Hsu et al., 2019), where the concentration parameter $\alpha$ controls heterogeneity. Small $\alpha$ produces highly skewed local distributions (clients dominated by few classes), while large $\alpha$ yields nearly uniform, IID-like splits ($\alpha \in \{0.1, 0.3, 0.5, 1.0\}$).

**Table 2:** Test accuracy (%) comparison across methods and datasets in a high heterogeneity setting ($\alpha = 0.1$).

| Method | CIFAR-10 | SVHN | MNIST |
|---|---|---|---|
| Random | 56.25 | 65.27 | 75.79 |
| Entropy | 64.23 | 85.31 | 87.63 |
| BADGE | 61.01 | 77.50 | 70.78 |
| Core-Set | 64.21 | 82.43 | 89.59 |
| Noise Stability | 60.04 | 81.71 | 80.53 |
| LoGo | 58.22 | 80.84 | 90.52 |
| KAFAL | 55.57 | 62.32 | 76.68 |
| FEAL | 57.08 | 69.98 | 72.72 |
| AHFAL (Ours) | **66.15** | **85.61** | **92.83** |

**Models and Training.** Our primary backbone is ResNet-8, trained locally for five epochs per communication round with aggregation via FedAvg. We also report results with MobileNetV2 to demonstrate robustness across architectures. Each experiment begins with 10% of the training set labeled at random. In every subsequent active learning cycle, clients add an additional 5% of labeled data according to the sampling strategy, and train for 100 communication rounds under FedAvg before the next cycle begins. All experiments are repeated with three random seeds, and we report mean accuracy with standard deviation. Further dataset-specific training details and hyperparameters are provided in the Appendix.

**Baselines.** We compare against ten baselines. First, *centralized methods* (local-only): Entropy (Holub et al., 2008), BADGE (Ash et al., 2019), Noise Stability (Li et al., 2024), Core-Set (Sener & Savarese, 2018), and Random. Second, *decentralized methods* (global-aware): KAFAL (Cao et al., 2023), LoGo (Kim et al., 2023), and FEAL (Chen et al., 2024).

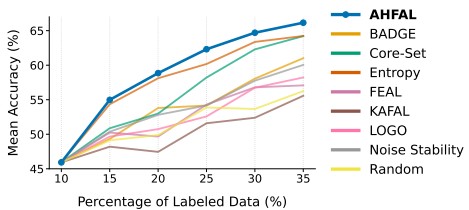

**Method hyperparameters.** AHFAL is implemented with the default threshold $\tau = 12$ (this is a threshold of class count variances standard deviation). We report a sensitivity study for $\tau$ on CIFAR-10 in the Appendix. The same threshold $\tau = 12$ is then used for all other datasets and heterogeneity settings, without additional setting-specific fine-tuning, and is found to work robustly. AHFAL consistently matches or outperforms baselines, demonstrating robust performance across datasets.

**Figure 7:** AHFAL offers state-of-the-art performance over centralized and decentralized active learning methods on CIFAR-10. For $\alpha = 0.1$ (high data heterogeneity), AHFAL is clearly superior across the board, at every active learning cycle.

### 7.2 PERFORMANCE COMPARISON

AHFAL consistently outperforms all centralized and decentralized baselines across datasets, heterogeneity levels, and architectures.

**Across Heterogeneity Levels.** Table 1 reports test accuracy for $\alpha \in \{0.1, 0.3, 0.5, 1.0\}$ for the CIFAR-10 dataset. Under strong heterogeneity ($\alpha = 0.1$), AHFAL achieves the highest accuracy (66.15%), exceeding the best centralized method (Entropy, 64.23%) and all federated methods. As heterogeneity decreases, baseline performance converges, yet AHFAL maintains a consistent margin over all competitors, demonstrating robustness across the entire spectrum from highly skewed to near-IID settings.

**Table 3:** Test accuracy (%) comparison across methods and datasets in a low heterogeneity setting ($\alpha = 1.0$).

| Method | CIFAR-10 | CIFAR-100 | SVHN | MNIST |
|---|---|---|---|---|
| Random | 77.71±0.44 | 43.32±0.25 | 92.68±0.35 | 97.96±0.10 |
| Entropy | 80.16±0.47 | 42.94±0.17 | 93.85±0.10 | 98.63±0.00 |
| BADGE | 77.87±0.26 | 41.71±0.25 | 92.30±0.21 | 97.90±0.00 |
| Core-Set | 79.00±0.41 | 43.98±0.30 | 93.13±0.17 | 98.61±0.00 |
| Noise Stability | 78.54±0.17 | 42.98±0.10 | 92.84±0.00 | 98.53±0.00 |
| LoGo | 79.06±0.72 | 43.93±0.91 | 93.56±0.00 | 98.41±0.00 |
| KAFAL | 79.25±0.72 | 43.46±0.10 | 93.90±0.22 | 98.37±0.00 |
| FEAL | 78.93±0.40 | 42.23±0.78 | 94.21±0.28 | 98.55±0.00 |
| AHFAL (Ours) | **79.82±0.39** | **44.03 ± 0.28** | **94.35±0.17** | **98.54±0.10** |

**Across Datasets.** Table 2 shows results on CIFAR-10, SVHN, and MNIST with $\alpha = 0.1$. AHFAL achieves the best accuracy in all cases. At lower heterogeneity ($\alpha = 1.0$), shown in Table 3, AHFAL matches or surpasses the strongest baselines on CIFAR-10, SVHN, MNIST, and CIFAR-100.

These results confirm that AHFAL adapts effectively to different datasets, including both simple (MNIST) and more challenging (CIFAR-100) benchmarks.

**Across architectures.** Table 4 compares performance on CIFAR-10 with $\alpha = 0.1$ using ResNet-8 and MobileNetV2. AHFAL outperforms all baselines on both architectures, achieving $77.68\%$ on MobileNetV2 compared to $76.05\%$ for Core-Set, the strongest baseline. This demonstrates that AHFAL's benefits are not architecture-specific.

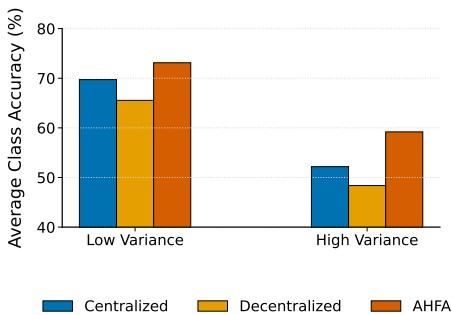

Across datasets, heterogeneity regimes, we note that AHFAL remains state-of-the-art (within error bounds) in these low heterogeneity settings, as well as being clearly superior in the high heterogeneity settings as shown in the paper (Table 1). These results confirm the promise of adaptive class-wise sampling as a consistent and effective strategy for federated active learning.

**Figure 8:** On CIFAR-10 for $\alpha = 0.1$, AHFAL outperforms prior centralized and decentralized active learning on average, while reducing performance discrepancy between high, low variance classes.

Figure 7 shows the accuracy curves across active learning rounds on the CIFAR-10 dataset, demonstrating that AHFAL not only achieves higher final accuracy in the high heterogeneity setting ($\alpha = 0.1$) but also exhibits better performance across all labeling budgets.

**Class-Specific Performance.**

Figure 8 shows results on CIFAR-10 under high client heterogeneity ($\alpha = 0.1$). Existing centralized and decentralized methods exhibit substantial performance gaps between high- and low-variance classes, averaging $17.54\%$ and $17.17\%$, respectively.

AHFAL improves performance across both class types, with particularly strong gains for high-variance classes, while reducing discrepancy to $13.93\%$. These findings align with our motivational analysis (Figures 2 and 3) and confirm that AHFAL reduces class-level disparities while simultaneously improving overall accuracy.

**Table 4:** Test accuracy (%) comparison across methods and architectures ($\alpha = 0.1$).

| Method | ResNet-8 | MobileNetV2 |
|---|---|---|
| Random | 56.25±3.73 | 66.64±0.91 |
| Entropy | 64.23±3.48 | 73.28±0.35 |
| BADGE | 61.01±1.37 | 73.27±1.33 |
| Core-Set | 64.21±1.20 | 76.05±0.97 |
| Noise Stability | 60.04±3.93 | 75.66±1.93 |
| LoGo | 58.22±4.98 | 70.33±2.86 |
| KAFAL | 55.57±4.75 | 65.38±6.35 |
| FEAL | 57.08±1.98 | 66.84±1.94 |
| **AHFAL (Ours)** | **66.15±0.94** | **77.68±1.35** |

### 7.3 PRIVACY–UTILITY TRADE-OFF

We next assess the empirical impact of the privacy mechanisms introduced in Section 6.3. Specifically, we evaluate AHFAL under local differential privacy, where each client perturbs its histogram with Laplace noise calibrated to different privacy budgets $\varepsilon$.

**Table 5:** AHFAL is robust across local differential privacy constraints. The total privacy budget $\varepsilon$ is distributed equally across active learning cycles.

| Algorithm | Total privacy budget $\varepsilon$ | Privacy budget per cycle | Accuracy (%) |
|---|---|---|---|
| AHFAL | 5 (strong privacy) | 1 | 65.70 |
| AHFAL | 10 (moderate privacy) | 2 | 65.74 |
| AHFAL | – | – | 66.15 |
| Best baseline | – | – | 64.23 |

Results in Table 5 on CIFAR-10 show that AHFAL maintains strong performance even under strict privacy constraints. Accuracy decreases only modestly compared to the non-private variant and consistently remains above the strongest baselines. This favorable trade-off arises because noise is applied to class histograms, which only indirectly affects learning, in contrast to noise injected directly into raw data or gradients. We also note that secure aggregation (Section 6.3) will incur negligible overhead in this setting, as only low-dimensional class histograms are exchanged. In combination, these findings demonstrate that AHFAL can be deployed under strong privacy guarantees without sacrificing its effectiveness.

### 7.4 COMMUNICATION AND COMPUTATION COSTS

**Communication overhead.** The extra communication from sharing class histograms is negligible compared to model transmission, which dominates FL. In our CIFAR-10 setup, each client sends a 10-dimensional class distribution vector (4 bytes per float, i.e., 40 bytes) per round, or 400 bytes total for 10 clients, versus $\approx 3.12$ MB of model parameters per round ($\sim 311$ KB per client), i.e.,

only $\approx 0.013\%$ additional overhead. Histogram size scales as $\mathcal{O}(KC)$ for $K$ classes and $C$ clients, but remains small even in extreme cases: for $K = 100$, $C = 1000$, the total histogram payload is $\approx 400$ KB, still below $0.1\%$ of typical model transmission costs (hundreds of MB). Thus, AHFAL's communication footprint is negligible even at large scale.

**Computation overhead.** AHFAL's computational profile is similar to decentralized methods such as KAFAL and is dominated by neural network inference. For each candidate sample, AHFAL performs two forward passes (local and global) to compute hybrid pseudo-labels, matching KAFAL's dual evaluations for KL-divergence–based scoring; the per-sample cost is of the same order as other baselines. Histogram aggregation and class-balancing add only $\mathcal{O}(KC)$ operations and contributed less than $0.1$ ms per round in our implementation. End-to-end, AHFAL's runtime is comparable to (and only marginally higher than) KAFAL, and significantly faster than methods like BADGE that incur additional clustering and gradient-computation overhead, making the extra $2\times$ forward pass a computationally efficient trade-off for improved class balancing under non-IID data.

### 7.5 Ablation Study

To evaluate the contribution of each component, we conduct an ablation study of AHFAL. Table 6 presents these results, evaluated on CIFAR-10 under $\alpha = 0.1$.

The ablation results confirm that each component contributes to AHFAL's performance (Table 6, row 1). Removing adaptive selection (i.e. enforcing a purely centralized approach to uncertainty estimation using only the local model) results in a minor performance degradation (Table 6, row 2). Removal of the class

**Table 6:** Ablation study on CIFAR-10 ( $\alpha = 0.1$).

| Method Variant | Accuracy (%) |
|---|---|
| AHFAL (Full) | 66.15 |
| AHFAL w/o centralized vs decentralized toggling | 65.89 |
| AHFAL w/o toggling, w/o class balance (entropy) | 64.23 |

balancing scheme that focuses on reducing inter client heterogeneity leads to further worsening of performance (Table 6, row 3). The method now degenerates to entropy-based centralized sampling (more analysis in supplement).

## 8 Discussion

We present Adaptive Hybrid Federated Active Learning (AHFAL), a framework for understand active learning in federated settings. AHFAL introduces the idea of leveraging client-side class histograms to estimate inter-client variance and to guide sample selection. This enables sampling policies that adapt at the class level—an approach not explored in prior work.

This contribution is significant because heterogeneity in federated learning is rarely uniform: some classes are broadly distributed, others concentrate on few clients. Existing methods ignore such variation, applying uniform strategies across all classes. By explicitly adapting to class-specific heterogeneity, AHFAL improves accuracy and label efficiency across datasets, heterogeneity regimes, and model architectures. Beyond empirical gains, AHFAL reframes federated active learning around heterogeneity management rather than sample-level heuristics. This has important implications for domains where annotation is especially costly. In medical collaborations, for example, labeling requires scarce expert time and is particularly limited for rare conditions. By prioritizing samples from underrepresented classes and balancing global and local querying, AHFAL can reduce the labeling workload for clinicians while improving overall model quality.

**Limitations and Future Work** The current framework assumes static distributions across active learning rounds; extending AHFAL to handle evolving client data remains an open challenge. Although our evaluation focuses on image classification, the principles of AHFAL could be extended to regression, structured prediction, and sequence modeling, provided suitable variance metrics and selection strategies are developed. Exploring these directions would further broaden the scope and impact of AHFAL. Combining AHFAL with recent advances in self-supervised learning has the potential to further reduce labeling requirements in collaborative settings.

## Reproducibility Statement

All theoretical assumptions are stated and numbered in Appendix C. The full method is specified in Sections 6 and 7.1, with algorithmic pseudocode in Appendix I and implementation details in Appendix J. We provide source code as supplementary material with fixed random seeds that reproduce all reported tables and figures.

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

## A    APPENDIX

This appendix is organized as follows:

1. Section B discusses the relevance and importance of the academic direction of this work.

2. Section C discusses the mathematical details of the theoretical analysis.

3. Section D analyzes additional context in terms of our comparision with Cao et al. (2023).

4. Section E discusses data heterogeneity and class-aware selection in further detail.

5. Section F discusses additional experimental results, under varying data heterogeneity

6. Section G discusses experimental results under varying labeling budgets

7. Section H discusses the global distribution alignment strategy.

8. Section I presents the overall proposed AHFAL algorithm.

9. Section J introduces further implementation details.

10. Section K discusses our included code.

11. Section L discusses LLM usage to write this manuscript.

12. Section M discusses additional experiments for empirical motivation of AHFAL.

## B  RELEVANCE OF THIS WORK

This work introduces Adaptive Hybrid Federated Active Learning (AHFAL), a novel approach to federated active learning that addresses critical limitations in heterogeneous data environments. While existing FAL methods have predominantly focused on informative sample selection strategies, we make the key observation that such approaches fundamentally fail in federated settings characterized by significant data heterogeneity. Our analysis reveals that mitigating heterogeneity-related challenges is more crucial than optimizing sample informativeness in these distributed environments. To address this gap, we present a principled yet practical method that prioritizes heterogeneity mitigation as a core component of federated active learning. We anticipate that our analysis and proposed approach will establish heterogeneity-aware design as an essential paradigm for developing robust FAL methods that maintain effectiveness across diverse data distribution scenarios.

We also observe that when compared to traditional baselines (other FAL methods), the proposed method demonstrates clear superiority (see Table 1, main paper). However, our analysis reveals that centralized methods also warrant comparison in this context. AHFAL proves to achieve state-of-the-art performance across a comprehensive range of heterogeneity configurations, establishing its effectiveness relative to both decentralized and centralized methods.

## C  THEORETICAL FOUNDATIONS

To explain the empirical findings in Section 4, we study entropy estimation under client heterogeneity. Our goal is to relate classwise performance to inter-client variance for each class $c$, comparing decentralized (global) and centralized (local) scoring.

We view acquisition scoring as *estimating the Bayes predictive entropy*

$$H^\star(x) \triangleq H\big(p^\star(\cdot \mid x)\big) = -\sum_{y=1}^{C} p^\star(y \mid x) \log p^\star(y \mid x),$$

where $p^\star(y \mid x)$ is the population conditional. Let $\hat{H}_c^L(x)$ and $\hat{H}_c^G(x)$ denote the predictive entropies from the client-local model $f_{\theta_i}^L$ and the federated/global model $f_\theta^G$, respectively, when the (pseudo)label of $x$ is class $c$. We analyze mean-squared error (MSE) with respect to $H^\star(x)$, averaging over $x \sim \mathcal{D}_{i,c}$ (client $i$'s class-$c$ pool).

### C.1  ENTROPY ESTIMATION UNDER HETEROGENEITY

Fix a client $i$ and class $c$. Using a bias–variance decomposition,

$$\hat{H}_c^L(x) = H^\star(x) + b_{i,c}^L(x) + \varepsilon_{L,i,c}(x), \tag{5}$$

$$\hat{H}_c^G(x) = H^\star(x) + \beta_c(x) + \varepsilon_{G,c}(x), \tag{6}$$

where $b_{i,c}^L(x)$ is the client–class specific bias (e.g., from limited local data or local optimizer noise), $\beta_c(x)$ is a class-specific bias induced by cross-client imbalance, and $\varepsilon_.$ are zero-mean fluctuations. Define

$$b_{i,c}^L \triangleq \mathbb{E}_x[\hat{H}_c^L(x) - H^\star(x)], \quad \beta_c \triangleq \mathbb{E}_x[\hat{H}_c^G(x) - H^\star(x)],$$

$$V_L \triangleq \mathrm{Var}_x(\hat{H}_c^L), \quad V_G \triangleq \mathrm{Var}_x(\hat{H}_c^G), \quad \rho \triangleq \mathrm{Cov}_x(\hat{H}_c^L, \hat{H}_c^G).$$

Let $\sigma_c$ be the cross-client standard deviation of the class-$c$ proportions.

**High-variance classes.** When $\sigma_c$ is large (class $c$ concentrated on few clients), the global model aggregates updates from many clients with sparse exposure to $c$, inducing a non-negligible $|\beta_c| > 0$. If client $i$ is *rich* in class $c$ (large $n_{i,c}$), then $b_{i,c}^L \approx 0$ and $V_L$ is small, so $\mathrm{MSE}(\hat{H}_c^L) \ll \mathrm{MSE}(\hat{H}_c^G)$; local dominates.

**Low-variance classes.** When $\sigma_c$ is small (class $c$ well spread), both estimators are approximately unbiased ($b_{i,c}^L \approx 0$, $\beta_c \approx 0$), and combining them can reduce variance.

**Client-poor case.** If client $i$ is *poor* in class $c$ (small $n_{i,c}$), then $b_{i,c}^L$ and $V_L$ can be large even if $\sigma_c$ is high; borrowing strength from the global estimator can still reduce MSE. This motivates using both global $\sigma_c$ and local $n_{i,c}$ (or a proxy) in the rule.

## C.2 Variance reduction via optimal ensemble

Consider $\hat{H}_c^{(\lambda)}(x) = \lambda \hat{H}_c^L(x) + (1-\lambda)\hat{H}_c^G(x)$ with $\lambda \in [0, 1]$. Its MSE is

$$\text{MSE}\big(\hat{H}_c^{(\lambda)}\big) = \big(\lambda b_{i,c}^L + (1-\lambda)\beta_c\big)^2 + \lambda^2 V_L + (1-\lambda)^2 V_G + 2\lambda(1-\lambda)\rho. \quad (7)$$

The minimizer is

$$\lambda^\star = \frac{V_G - \rho + \beta_c(\beta_c - b_{i,c}^L)}{V_L + V_G - 2\rho + (b_{i,c}^L - \beta_c)^2} \quad \text{clipped to } [0, 1]. \quad (8)$$

**Special case.** If both are unbiased ($b_{i,c}^L = \beta_c = 0$) and uncorrelated ($\rho = 0$), then $\lambda^\star = \frac{V_G}{V_L + V_G}$ and $\text{MSE}(\hat{H}_c^{(1/2)}) = \frac{1}{4}(V_L + V_G)$.

Heuristically, $V_L$ decreases with the local class count ($V_L \propto \frac{1}{n_{i,c}}$), while $|\beta_c|$ increases with cross-client imbalance (we assume $|\beta_c|$ is non-decreasing in $\sigma_c$). Then equation 8 implies: (i) for large $|\beta_c|$ (high $\sigma_c$), $\lambda^\star \to 1$ (favor local); (ii) for small $|\beta_c|$ and large $V_L$ (client-poor), $\lambda^\star$ moves toward hybrid/federated.

## C.3 Class partitioning

AHFAL chooses between *local* ($\lambda=1$) and a fixed *hybrid* ($\lambda=1/2$). Comparing equation 7 at $\lambda=1/2$ to local ($\lambda=1$) yields the following sufficient condition for hybrid to beat local when the local estimator is (approximately) unbiased ($b_{i,c}^L \approx 0$):

$$\text{MSE}\big(\hat{H}_c^{(1/2)}\big) < \text{MSE}\big(\hat{H}_c^L\big) \quad \Longleftarrow \quad \beta_c^2 < 3V_L - V_G - 2\rho. \quad (9)$$

Hence, when the global bias $\beta_c$ (increasing with $\sigma_c$) is too large relative to the local–global variance gap and covariance, pure local is optimal; otherwise, hybrid is preferable. Since $\beta_c$ is not directly observable, AHFAL uses $\sigma_c$ as a proxy via the monotonicity assumption.

**Assumptions and scope.** We assume predictive probabilities are bounded away from 0 and 1 (e.g., via temperature smoothing), ensuring continuity of $H(\cdot)$ and controlling variance. We also assume $|\beta_c|$ is non-decreasing in $\sigma_c$ under FedAvg-style aggregation (class imbalance skews the effective training distribution), and treat $\rho$ explicitly (we avoid assuming $\rho \geq 0$).

# D Additional Details on the Comparison with Cao et al. (2023)

The KAFAL algorithm (Cao et al., 2023) consists of two independent modules: (1) Knowledge-Specialized Active Sampling (KSAS), a query strategy that determines which samples to select from the unlabeled data, and (2) Knowledge-Compensatory Federated Update (KCFU), a local update mechanism that addresses class imbalance. To ensure a fair comparison, we isolated the effectiveness of different query strategies by comparing only the query strategies in the main paper, since the local update mechanism KCFU can be applied to all methods, including ours, to further enhance performance.

To verify the complementary effect of KCFU with our AHFAL method, we additionally evaluate KAFAL (+KCFU) and AHFAL (+KCFU). We conduct these experiments on CIFAR10 with $\alpha = 0.1$.

Table 7 reports final accuracy values and standard deviations across three trials. Figure 9 shows results across different labeling budgets. Adding KCFU yields consistent gains for both query strategies (KAFAL: +14.44%, AHFAL: +7.83%). AHFAL already significantly outperforms KAFAL without KCFU but still benefits from the additional

**Table 7:** AHFAL with Knowledge Compensatory Federated Update on CIFAR-10 ($\alpha = 0.1$).

| Method | Accuracy (%) |
|---|---|
| KAFAL Cao et al. (2023) (KSAS only) | 55.57 ±2.18 |
| KAFAL Cao et al. (2023) full (KSAS + KCFU) | 70.01 ±0.91 |
| AHFAL (ours) | 66.15 ±0.97 |
| AHFAL (ours) + KCFU | 73.98 ±0.92 |

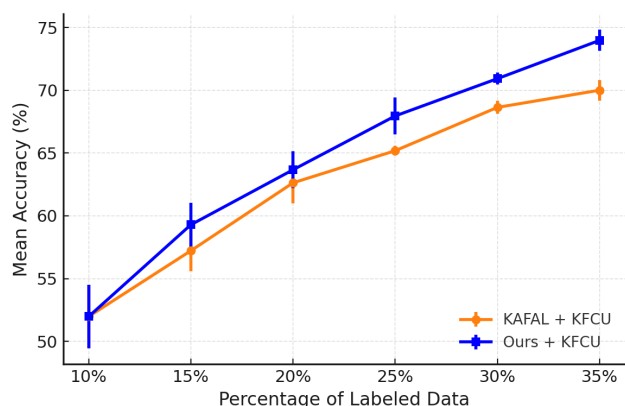

**Figure 9:** Comparison of KAFAL and AHFAL both with KFCU on cifar10 with $\alpha = 0.1$.

knowledge-compensatory update, confirming that our
sampling criterion and KCFU address distinct aspects of federated active learning. Even after equipping both methods with KCFU, our approach remains superior, outperforming KAFAL (+KCFU) by 3.97%, demonstrating that the improvements from our AHFAL method are complementary to those from knowledge compensation.

## E  DATA HETEROGENEITY AND CLASS-AWARE SELECTION

We first present a visual representation of data heterogeneity. Figure 10 depicts the class-frequency distributions across clients under three Dirichlet concentration parameters ($\alpha = 10.0, 0.5, 0.05$). Even at a fixed $\alpha$, we observe that classes are not distributed evenly—some classes exhibit high across-client variance (i.e., most of their samples reside on a single or very few clients) while others are low-variance and spread more evenly.

To exploit this structure, our heterogeneity-aware update first computes, for each class $c$, the empirical variance across clients. We then compare the standard deviation of each class against a threshold $\tau$. For classes whose variance exceeds $\tau$, we perform updates using only the local model: when a class is concentrated on few clients, global aggregation risks diluting its unique features, so pure local optimization avoids "noise" from unrelated data. Conversely, for classes with variance below $\tau$, we combine local and global model updates, since well-distributed classes benefit from the richer, aggregated representation. Figure 11 illustrates how $\tau$ governs per-class strategy selection under varying heterogeneity: high-variance classes use a local-only update, while low-variance classes employ a global-aware update. As $\alpha$ increases (heterogeneity decreases), more classes fall below the threshold and adopt the hybrid strategy. Furthermore, as active learning cycles progress and underrepresented classes accrue more labeled examples, their variances naturally decline, allowing additional classes to transition to global-aware updates

This thresholding approach proves robust across regimes. In the high-heterogeneity setting ($\alpha = 0.1$, Figure 11), most classes exceed $\tau$ initially, so most classes start by using only the local model to select samples. As our active learning cycles progress and more samples of underrepresented classes are labeled, their per-class variances decrease; consequently, additional classes cross below $\tau$ and begin to incorporate global knowledge as well. Under the near-IID regime ($\alpha = 0.5$), many classes already lie under the threshold at the outset, yielding rapid hybrid updates for the majority of classes.

The threshold $\tau = 12$ was determined empirically. For example, sensitivity analysis on CIFAR-10 ($\alpha = 0.1$) shows: $\tau = 10$ (64.60%), $\tau = 12$ (65.51%), $\tau = 15$ (62.21%). Performance varies by only about 3% across this range, indicating reasonable robustness to threshold selection.

As a result of the proposed approach, we find that AHFAL shows state of the art performance across a range of data heterogeneities, ranging from high to low data heterogeneities. Table 1 (main paper) highlights this superior performance.

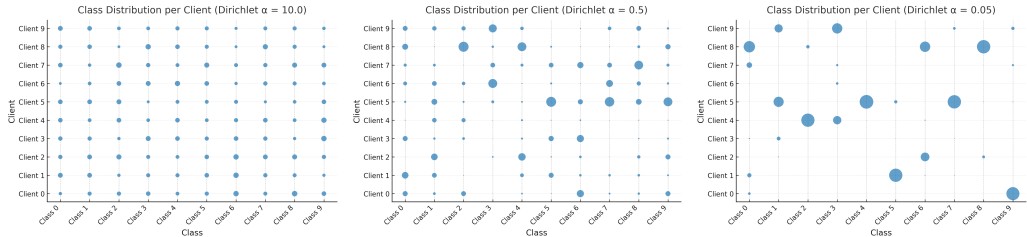

**Figure 10:** A visualization of a Dirichlet partition with $\alpha$ values of 10.0, 0.5 and 0.05, across 10 clients and 10 classes. A lower Dirichlet parameter leads to higher data heterogeneity between clients and classes.

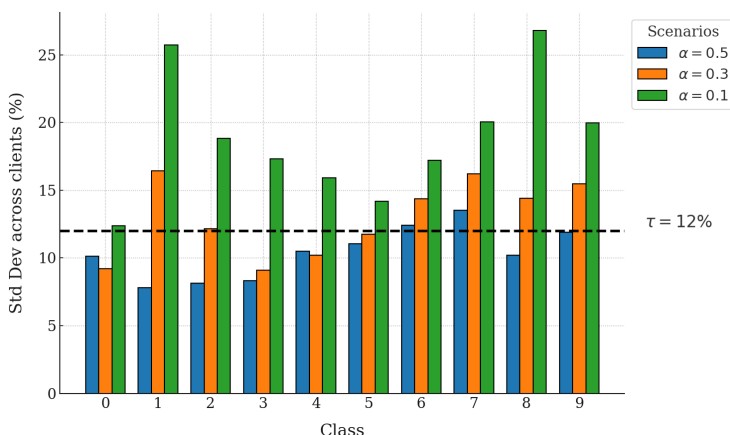

**Figure 11:** Illustration of how AHFAL adaptively uses either a centralized strategy (if std. deviation $> \tau$) or a decentralized strategy (if std. deviation $< \tau$).

# F FURTHER EXPERIMENTAL RESULTS UNDER VARYING DATA HETEROGENEITY

**Table 8:** Test accuracy (%) comparison across methods and data heterogeneities on **MNIST** (left) and **SVHN** (right).

| Method | $\alpha = 0.1$ | $\alpha = 0.5$ | $\alpha = 1.0$ | Method | $\alpha = 0.1$ | $\alpha = 0.5$ | $\alpha = 1.0$ |
|---|---|---|---|---|---|---|---|
| Random | 75.79 | 98.88 | 97.96 | Random | 65.27 | 90.89 | 92.68 |
| Entropy | 87.63 | 99.20 | 98.63 | Entropy | 85.31 | 92.15 | 93.85 |
| BADGE | 70.78 | 98.81 | 97.90 | BADGE | 77.50 | 90.57 | 92.30 |
| Core-Set | 89.59 | 99.18 | 98.61 | Core-Set | 82.43 | 91.08 | 93.13 |
| Noise Stability | 80.53 | 99.02 | 98.53 | Noise Stability | 81.71 | 90.50 | 92.84 |
| LoGo | 90.52 | 99.21 | 98.41 | LoGo | 80.84 | 91.59 | 93.56 |
| KAFAL | 76.68 | 99.02 | 98.37 | KAFAL | 62.32 | 91.51 | 93.90 |
| FEAL | 72.72 | 99.06 | 98.55 | FEAL | 69.98 | 92.08 | 94.21 |
| **AHFAL (Ours)** | **92.83** | **99.16** | **98.54** | **AHFAL (Ours)** | **85.61** | **91.80** | **94.35** |

# G FURTHER EXPERIMENTAL RESULTS UNDER VARYING LABELING BUDGETS

Table 9 evaluates the impact of varying the labeling regime. In the low-budget regime, where we halve both the initial labeled pool and the labels queried per round (up to 15% labeled data), AHFAL achieves 59.42%, outperforming all other methods. In the high-budget regime, where we double the

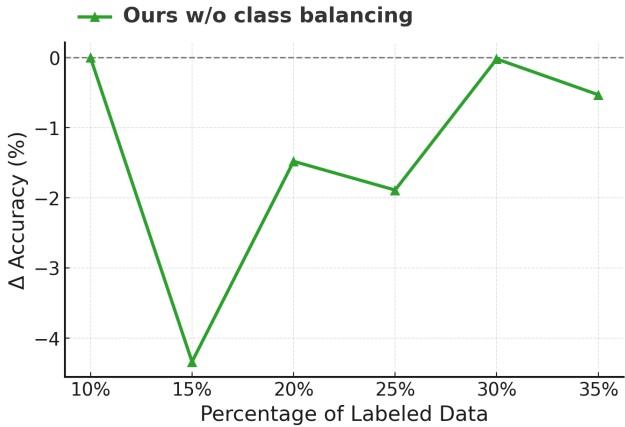

**Figure 12:** Comparison of AHFAL with and without Class Balancing Selection Strategy on $\alpha = 0.1$, in terms of accuracy.

initial labeled pool and per-round query size (up to 60% labeled data), AHFAL again surpasses all baselines. These results show that AHFAL remains consistently superior across labeling schedules.

## H    GLOBAL DISTRIBUTION ALIGNMENT

AHFAL employs a representation-ratio-based balancing strategy that prioritizes underrepresented classes to align local client distributions with the global data distribution (See Algorithm 1). The target global distribution $D_{global}(c)$ represents the estimated true proportion of class $c$ across all clients in the federation. This serves as the ideal reference distribution toward which each client should strive. During labeling, more budget is devoted to classes that are currently underrepresented, a moderate share goes to those with some underrepresentation, and the remainder is used for adequately represented classes. This dynamic allocation guides each client's labeled set toward the global target distribution.

**Table 9:** Test accuracy (%) comparison across labeling regimes ($\alpha = 0.1$).

| Method | Budget Low | Budget High |
|---|---|---|
| Random | $49.04 \pm 2.33$ | $60.03 \pm 4.26$ |
| Entropy | $57.47 \pm 2.88$ | $63.10 \pm 0.74$ |
| BADGE | $49.69 \pm 3.47$ | $59.23 \pm 3.23$ |
| Core-Set | $52.79 \pm 1.94$ | $62.65 \pm 2.45$ |
| Noise Stability | $47.06 \pm 3.57$ | $63.99 \pm 1.14$ |
| LoGo | $48.38 \pm 4.05$ | $61.66 \pm 3.86$ |
| KAFAL | $48.39 \pm 2.96$ | $59.04 \pm 1.24$ |
| FEAL | $51.15 \pm 3.08$ | $58.24 \pm 2.34$ |
| AHFAL (Ours) | $\mathbf{59.42 \pm 1.92}$ | $\mathbf{66.30 \pm 1.25}$ |

To isolate the impact of class balancing, we also evaluated AHFAL without this mechanism. Figure 12 shows that removing class balancing leads to a noticeable drop in accuracy during the first round, with performance gradually recovering over subsequent cycles. Owing to significant data heterogeneity, there is a natural limit to how closely an individual client can match the global distribution and most of the alignment is achieved within the initial labeling cycles in high heterogeneity settings.

## I    ALGORITHM

We also include an explicit algorithmic description of AHFAL in the supplement in the form of algorithmic pseudocode in Algorithm 1. This includes the local model training, global federated learning, as well as 3 key steps of AHFAL sampling: first, the global class distribution, class variances and class partitioning into low and high variance groups is calculated and broadcasted by the server. Then, the hybrid uncertainty scoring is carried out as a function of class variance. Finally, class-aware sample allocation is carried out based on the uncertainty scores for all unlabeled samples.

## J    IMPLEMENTATION DETAILS

We implement AHFAL with the default threshold $\tau = 12.0$. In each communication round, every client trains its local model for 5 epochs before model aggregation via FedAvg, repeated for 100

---

**Algorithm 1:** AHFAL: Adaptive Hybrid Federated Active Learning

---

**Input:** Clients $1{:}N$; initial labelled sets $\{\mathcal{L}_i^{(0)}\}$ (10%); unlabelled pools $\{\mathcal{U}_i^{(0)}\}$; initial global model $\theta^{(0)}$; per-round labelling budget $B$; local epochs $E$; total rounds $R$; variance threshold $\tau$.

**Output:** Final global model $\theta^{(R)}$.

1 **for** $r \leftarrow 0$ **to** $R-1$ **do**              `// federated rounds`

  /* **Local training**                     */

2 **for** *each client* $i = 1{:}N$ **do in parallel**

3   Train $f_{\theta_i}^{\mathrm{L}}$ on $\mathcal{L}_i^{(r)}$ for $E$ epochs

4   Send updated weights $\theta_i$ to server

5 $\theta^{(r+1)} \leftarrow \text{FEDAVG}\big(\{\theta_i\}_{i=1}^{N}\big)$

6 Broadcast $\theta^{(r+1)}$ to all clients

  /* **AHFAL sampling**                     */

7 Clients compute $\mathbf{p}_i$ locally and send to server

8 Server returns $(\bar{\mathbf{p}}, \boldsymbol{\sigma})$

9 Define $\mathcal{C}_{\mathrm{low}}, \mathcal{C}_{\mathrm{high}}$ using variance threshold $\tau$

10 **for** *each client* $i = 1{:}N$ **do in parallel**

11  **for** $x \in \mathcal{U}_i^{(r)}$ **do**

12   Compute $H(x)$ via entropy calculation

13  Determine class budgets $\mathbf{b}$ via budget allocation

14  $\mathcal{S}_i \leftarrow$ top-$b_c$ samples per class $(|\mathcal{S}_i| = B)$

15  Query oracle for labels of $\mathcal{S}_i$

16  $\mathcal{L}_i^{(r+1)} \leftarrow \mathcal{L}_i^{(r)} \cup \mathcal{S}_i$

17  $\mathcal{U}_i^{(r+1)} \leftarrow \mathcal{U}_i^{(r)} \setminus \mathcal{S}_i$

---

rounds. For MNIST, given its lower complexity, we run 10 local epochs and 10 communication rounds. CIFAR-10 and CIFAR-100 experiments are run until 35% of samples are labeled, and 40% for the other datasets. Training uses SGD with learning rate 0.1, batch size 128, and momentum 0.9. All results are averaged over three random seeds for statistical significance. Experiments were executed on NVIDIA T4, A100, and H100 GPUs.

# K  CODE

We include the reference code implementation, along with a README file that runs through details on how to run the code, as part of the submission files along with the supplementary material.

# L  LLM USAGE

LLM assistance has been used to refine the writing of some parts of this manuscript.

# M  SUPPLEMENTARY EXPERIMENTS FOR EMPIRICAL MOTIVATION

We conduct additional empirical experiments on the SVHN (Figure 13 14) and MNIST (Figure 15 16) datasets (to supplement results from Section 4 of the main paper). Specifially, we aim to validate Key Finding 1 and 2 across these additional datasets to further solidify our motivation of AHFAL.

We find that our observations from CIFAR-10 hold true across both MNIST and SVHN, for Key Findings 1 and 2. Namely, aggregate heterogeneity drives the centralized-decentralized trade-off, and class-wise variance explains the crossover point.

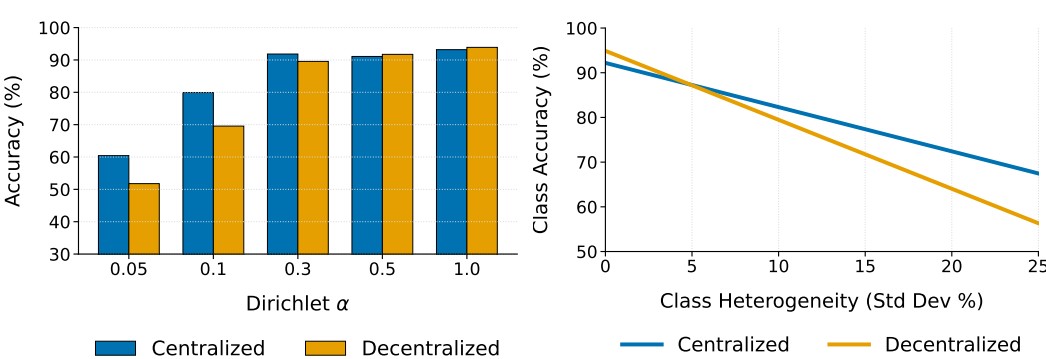

**Figure 13: Aggregate heterogeneity tradeoff (SVHN).** Decentralized strategies excel when client distributions are similar (large $\alpha$), while centralized methods dominate under strong heterogeneity (small $\alpha$).

**Figure 14: Class-wise variance explains the crossover (SVHN).** Classes with high $CV_c$ favor centralized sampling, while low-variance classes benefit from decentralized selection. Each line is a least-squares fit.

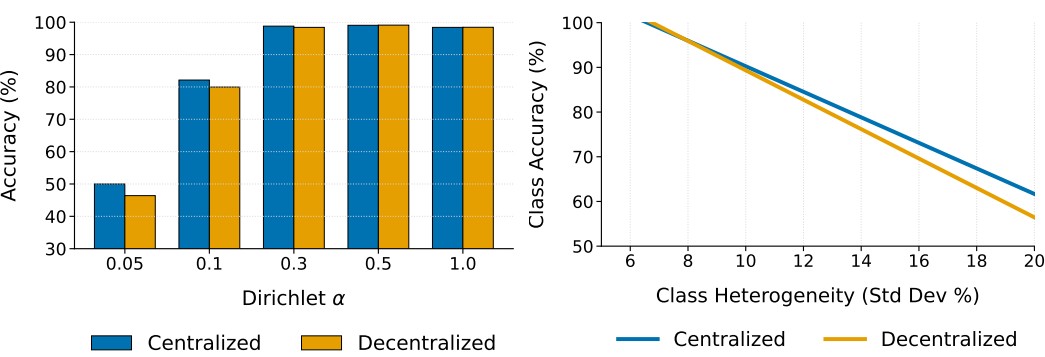

**Figure 15: Aggregate heterogeneity tradeoff (MNIST).** Decentralized strategies excel when client distributions are similar (large $\alpha$), while centralized methods dominate under strong heterogeneity (small $\alpha$).

**Figure 16: Class-wise variance explains the crossover (MNIST).** Classes with high $CV_c$ favor centralized sampling, while low-variance classes benefit from decentralized selection. Each line is a least-squares fit.

