# OpenReview forum: "Federated Active Learning via Class-adaptive Local–Global Balancing"
_ICLR.cc/2026/Conference — Submitted to ICLR 2026_

### Official Review · Reviewer_vAKh · 2025-10-28

**Soundness:** 3
**Presentation:** 3
**Contribution:** 3
**Rating:** 4
**Confidence:** 3

**Summary:**

The paper introduces Adaptive Hybrid Federated Active Learning, abbreviated as AHFAL, a method that adaptively chooses between centralized and decentralized active learning depending on the class specific distribution of data across clients. The central idea is that the performance of federated active learning depends on how unevenly classes are distributed among clients. The authors show that centralized querying is more effective when heterogeneity is high, while decentralized querying performs better when data are more uniformly distributed. AHFAL measures class wise variance to decide which strategy to use per class. The framework is designed to integrate naturally with standard federated learning pipelines and includes privacy considerations through local differential privacy and secure aggregation.

**Strengths:**

The work identifies a fundamental factor in federated active learning, namely that class wise data heterogeneity determines which strategy performs best. This insight is novel and well supported by both analysis and experiments.

The theoretical part connects bias and variance of entropy estimators to class variance, which gives an intuitive explanation of when local or global models are preferable.

The proposed method is conceptually simple, yet general enough to apply to different datasets and model architectures. The paper also considers privacy constraints and proposes reasonable mechanisms to preserve data confidentiality.

It is clearly written, logically structured, and the figures help convey the main findings.

**Weaknesses:**

The evaluation of the paper does not fully follow best practices in active learning research. As highlighted by Lüth, Bungert, Klein, and Jaeger (2023) [1], several recurring pitfalls appear in the design and evaluation of active learning methods, some of which are present here.
* The evaluation lacks diversity in data distribution settings. The paper does not include experiments on imbalanced datasets, although such conditions are central to realistic federated learning scenarios.
* The evaluation settings are narrow. The paper fixes the query size at five percent and reports results only after seven active learning rounds for CIFAR10 and CIFAR100, and eight rounds for the other datasets. A broader evaluation using different labeling budgets and query schedules would provide a clearer picture of method behavior. In addition, other dataset partitioning schemes could be applied, such as a variant of McMahan et al. (2017) [2], where the level of heterogeneity is controlled by the shard size and the number of shards per client.
* The paper overlooks the role of classifier configuration. There is no distinction between development datasets, used for hyperparameter tuning, and roll out datasets, used for final evaluation. The threshold parameter Tau should be selected on some datasets and tested on others to verify generalization.
* The experiments do not consider alternative training paradigms. Only models trained from scratch are used, without testing pre-trained backbones or other initialization strategies that are standard in current active learning research.

The paper also lacks quantitative discussion of computational and communication overhead introduced by sharing class histograms and computing hybrid uncertainty scores.

Table 2 is hard to read.

[1] Lüth, Bungert, Klein, and Jaeger (2023) propose a systematic evaluation framework for active learning and analyze common pitfalls in AL literature.

[2] McMahan, B., Moore, E., Ramage, D., Hampson, S., & Agüera y Arcas, B. (2017). Communication-efficient learning of deep networks from decentralized data (AISTATS, JMLR W&CP Vol. 54). arXiv preprint arXiv:1602.05629. Retrieved from https://arxiv.org/abs/1602.05629

**Questions:**

How would AHFAL perform under imbalanced data distributions, for example when some classes are heavily underrepresented across clients?

Why was a fixed query size of five percent chosen, and how does the method behave under different query budgets or adaptive labeling schedules?

How does AHFAL perform using more than one dataset partitioning scheme, such as variants of the shard based partitioning proposed by McMahan et al. (2017)?

What is the performance with the same hyperparameters on 1-2 new dataset on which the hyperparameters were not tuned?

Instead of using entropy for sampling in Equation (3), have you considered alternative sampling strategies such as margin sampling [3]?

How does AHFAL perform with pre trained backbones, and if not, how might pre training affect the observed trade off between centralized and decentralized sampling?

How many datapoints do you have for Figure 3? The line looks straight, is it just 4 datapoints?

[3] Bahri, Dara; Jiang, Heinrich; Schuster, Tal; Rostamizadeh, Afshin (2022). Is margin all you need? An extensive empirical study of active learning on tabular data. arXiv preprint arXiv:2210.03822.

---

> ### Author Response · Authors · 2025-11-21
>
> We thank all reviewers for their careful reading and constructive feedback. We are encouraged that the central insight of the paper, that class-wise data heterogeneity is a fundamental driver of which federated active learning strategy works best and that the strategy should adapt at the class level, is viewed as novel and well supported by our analysis and experiments (R fNx8, R vAKh). We also appreciate the positive comments that the paper is well written, easy to follow, and clearly structured, with figures that help convey the main findings (R BM9g, R XryK, R vAKh).
>
> Reviewers further noted that AHFAL is simple and practical, relying only on low-dimensional class histograms that fit naturally into standard FL workflows while admitting privacy-preserving mechanisms (R fNx8, R vAKh). The method is seen as effective and fairly general, with strong performance improvements over both centralized and federated baselines across datasets and architectures, and a code release that supports reproducibility and follow-up work (R fNx8, R BM9g, R vAKh). We now respond point by point to specific questions.
>
> **Q1. The evaluation only considers balanced benchmarks and does not explicitly study imbalanced data distributions. In realistic federated settings, some classes can be heavily underrepresented across clients. How would AHFAL perform under such imbalanced conditions?**
>
> We appreciate the reviewer’s concern about realistic data imbalance in federated settings. Our evaluation in fact already includes a naturally imbalanced dataset: SVHN. Unlike CIFAR-10/100, the standard SVHN training set exhibits non-uniform digit frequencies (with majority classes appearing roughly 1.5–2× more often than minority ones), making it closer to real-world FL deployments. SVHN has also been used in prior FAL work (e.g., LoGo) specifically to study performance under such natural imbalance.
>
> In our experiments, AHFAL consistently outperforms all baselines on SVHN as well, indicating that the proposed class-adaptive mechanism remains effective under imbalanced class distributions.
>
>
> **Q2. The evaluation uses a fixed query size of 5%. This is a relatively narrow setup. Why was 5% chosen, and how does AHFAL behave under different label budgets or query schedules (e.g., smaller or larger total labeling budgets)?**
>
> We thank the reviewer for highlighting the importance of evaluating under varied labeling regimes. Our main experiments use a 5% query size per round to match prior FAL work (e.g., KAFAL), ensuring fair and reproducible comparisons under the same labeling protocol and total label budget.
>
> To directly address this concern, we performed additional experiments on CIFAR-10 ($\alpha = 0.1$), varying both the initial labeled fraction and per-round query size to study low- and high-budget regimes.
>
> ---
>
> **High-budget regime (up to 60% labeled).**
> We doubled both the initial labeled pool and the per-round query size, and continued querying until 60% of the data was labeled.
>
>
> ### **CIFAR-10, $ \alpha = 0.1$**
>
>
> | Strategy | Accuracy [%] | StdDev |
> | -------- | ------------ | ------ |
> | AHFAL    | **66.30**    | 1.25   |
> | Core-Set | 62.65        | 2.45   |
> | Entropy  | 63.10        | 0.74   |
> | FEAL     | 58.24        | 2.34   |
> | KAFAL    | 59.04        | 1.24   |
> | LoGo     | 61.66        | 3.86   |
> | Noise    | 63.99        | 1.14   |
> | Random   | 60.03        | 4.26   |
>
> AHFAL attains the highest accuracy in this regime, maintaining a clear margin over all baselines.
>
> ---
>
> **Low-budget regime (up to 15% labeled).**
> We halved both the initial labeled pool and the per-round query size, and stopped once 15% of the data was labeled.
>
>
> ### **CIFAR-10,  $\alpha = 0.1$**
>
>
> | Strategy | Accuracy [%] | StdDev |
> | -------- | ------------ | ------ |
> | AHFAL    | **59.42**    | 1.92   |
> | BADGE    | 49.69        | 3.47   |
> | Core-Set | 52.79        | 1.94   |
> | Entropy  | 57.47        | 2.88   |
> | FEAL     | 51.16        | 3.08   |
> | KAFAL    | 48.39        | 2.96   |
> | LoGo     | 48.38        | 4.05   |
> | Noise    | 47.06        | 3.57   |
> | Random   | 49.04        | 2.33   |
>
> In this low-budget setting, where label efficiency is most critical, AHFAL again achieves the best performance and shows the largest gains over the federated baselines.
>
> ---
>
> Overall, these results indicate that AHFAL remains consistently strong across low-, medium-, and high-budget labeling schedules, and its advantages are not an artifact of the 5% query size used in the main setup. We have incorporated these findings in the revised paper (Section 7.2).

---

> ### Author Response · Authors · 2025-11-21
>
> **Q3. The paper evaluates AHFAL only under Dirichlet-based non-IID partitions. Other schemes, such as shard-based partitioning in McMahan et al. (2017), can also control heterogeneity (via shard size and shards per client). How does AHFAL perform under these alternative partitioning schemes?**
>
>
> We thank the reviewer for this suggestion.
>
> In our current experiments, we primarily adopt the Dirichlet partitioning scheme, which has become the de facto standard for simulating controllable label skew in modern federated learning. It offers a continuous knob on heterogeneity (via α), makes it easy to sweep from highly skewed to near-IID regimes, and ensures that our results are directly comparable to a broad body of recent FL/FAL work that uses the same setup (many of which serve as baselines in this work, such as KAFAL, LoGo etc.).
>
> In addition, our evaluation already includes naturally imbalanced data via SVHN, whose label frequencies are not uniform in practice. This complements the synthetic Dirichlet splits and provides evidence that AHFAL behaves well under realistic, non-uniform class distributions, not just perfectly balanced benchmarks like CIFAR.
>
> We agree that shard-based partitioning à la McMahan et al. is a useful, complementary way to stress-test heterogeneity (especially extreme “few-class-per-client” scenarios). Due to the limited time in the rebuttal period, we have not yet incorporated a full shard-based sweep, but we are happy to add experiments with shard-based non-IID splits in the final version. These will follow the same protocol (same models, budgets, and baselines), and we expect the qualitative takeaway to remain: AHFAL’s adaptive handling of class-wise heterogeneity continues to provide gains precisely in the regimes where non-IID effects are strongest.
>
> **Q4. The threshold parameter \tau in the class-variance partitioning is not validated on held-out datasets. How well does AHFAL perform when \tau is applied to 1–2 additional datasets on which they were not tuned?**
>
> We appreciate the reviewer’s concern about hyperparameter configuration and generalization.
>
> Please refer to our concise sensitivity study on CIFAR-10 with $(\alpha = 0.1)$ in the Appendix:
>
> $\tau = 10$: 64.60\%, $\tau = 12$: 65.51\%, $\tau = 15$: 62.21\%.
>
>
>
> Performance changes by only ($\approx$ 3%) across this range, indicating that AHFAL is robust to the choice of $\tau$.
>
> This ($\tau = 12$) from this analysis, is then used for other datasets (CIFAR-100, SVHN, MNIST) and heterogeneity settings, without additional setting-specific fine tuning, and is found to work robustly. AHFAL still consistently matches or outperforms baselines in these settings, demonstrating that hyperparameters generalize across datasets.
>
> **Q5. The experiments only use models trained from scratch. Modern active learning usually relies on pre-trained backbones or stronger initializations. How would AHFAL behave with pre-trained models, and could pretraining change the trade-off you observe between centralized and decentralized sampling?**
>
> We thank the reviewer for this thoughtful point and fully agree that studying AHFAL with pre-trained backbones is an important direction for follow-up work.
> In this paper, our primary goal was to isolate the effect of client heterogeneity on federated active learning and to understand when local-only vs. federated-aware sampling is preferable.
>
> To keep this analysis clean and directly comparable to prior FAL work we build on, we adopted a controlled setting where all methods (centralized and federated) are trained from scratch under the same protocol. This avoids introducing additional confounders from heterogeneous pretraining schemes or backbone choices, and most importantly is consistent with how prior work is evaluated.
>
> Conceptually, we expect that using strong pre-trained backbones would:
> * Increase overall accuracy and reduce label budgets for all methods
> * Partially reduce variance in local estimators, especially in low-data regimes
> * The core mechanism we analyze, namely the bias and variance mismatch between local and global models induced by cross-client class imbalance, would still remain. As long as local and global models see systematically different mixtures of classes, pretraining will not fully eliminate the local–global trade-off; it will likely change its magnitude, not its existence.
>
> We see this work as deliberately focusing on the heterogeneity-driven trade-off in a clean setting, and we view extending AHFAL to pre-trained backbones and richer initialization strategies as a natural next step, which we plan to explore in future work.

---

> ### Author Response · Authors · 2025-11-21
>
> **Q6.  The paper lacks quantitative discussion of computational and communication overhead introduced by sharing class histograms and computing hybrid uncertainty scores.**
>
> We thank the reviewer for raising this point. We have now quantified both the communication and computational overhead of AHFAL and compared it to standard FAL baselines. We have incorporated these insights into the revised manuscript in Subsection 7.4.
>
> **Communication overhead**
>
> The additional communication introduced by AHFAL comes only from transmitting class histograms, which is negligible compared to standard model updates in FL.
>
> In our CIFAR-10 setup (10 classes, 10 clients):
>
>   * Per client, per cycle: a 10-dimensional float vector ($\mathbf{p}_i$)
>     ($\Rightarrow$) (10 $\times$ 4) bytes (= 40) bytes.
>   * Across 10 clients: (10 $\times$ 40 = 400) bytes per cycle.
>   * Model transmission per cycle: roughly (311 KB) per client
>     ($\Rightarrow$) ( $\approx$ 3.12 MB) across 10 clients.
>   * Overhead ratio: $\frac{400\ \text{bytes}}{3.12 \times 10^6\ \text{bytes}} \approx 0.013\%$.
>
>
> **Scaling to larger systems:**
>   With (C) classes and (N) clients, the histogram cost scales as (O(NC)).
>   Even with (N = 1000) clients and (C = 100) classes, each client sends
>   (100 $\times$ 4 = 400) bytes, for a total of $(\approx$ 400 KB ) per cycle.
>   Compared to typical FL model updates (tens–hundreds of MB for realistic models), this remains well below **0.1%** of the model communication cost.
>
> Thus, the extra communication required by AHFAL is several orders of magnitude smaller than the standard FL communication and remains negligible even at large scale.
>
> **Computational overhead**
>
> On the computation side, AHFAL is comparable to existing FAL methods and significantly cheaper than the heaviest baselines.
>
> * **Uncertainty scoring:**
>   AHFAL requires:
>
>   * one forward pass through the local model for all unlabeled samples (to get pseudo-labels and (H^L)), and
>   * a forward pass through the global model only for samples whose pseudo-labels belong to low-variance classes.
>     In the worst case, this corresponds to at most two forward passes per sample, which is similar to KAFAL, where KL divergences between local and global predictions are computed.
>
> * **Histogram and routing logic:**
>   Class-histogram aggregation and the ($\sigma_c$)-based partitioning contribute only (O(NC)) operations per round, which amounts to **< 0.1 ms** per cycle in our implementation; this is negligible relative to local training and inference over the unlabeled pool.
>
> * **Wall-clock measurements** (CIFAR-10, ($\alpha = 0.1$), averaged over 3 trials):
>
>   * Random: ~5h 24m
>   * Entropy: ~5h 29m
>   * Core-Set: ~5h 30m
>   * LoGo: ~5h 28m
>   * KAFAL: ~5h 27m
>   * FEAL: ~5h 24m
>   * Noise Stability: ~6h 24m
>   * **AHFAL:** ~5h 34m
>   * BADGE: ~7h 25m
>
> AHFAL’s runtime is within a few minutes of KAFAL, Entropy, and Core-Set, and substantially faster than BADGE, whose gradient-based, high-dimensional clustering incurs much higher computational cost. This empirically supports our claim that AHFAL’s extra logic is lightweight relative to the dominant forward/backward passes already present in FAL pipelines.
>
> We have added these communication and computation measurements, as well as the (O(NC)) scaling analysis, to the revised version (Section 7.4) to make the “lightweight overhead” claim precise and transparent for large-scale settings.
>
> **Q7. Table 2 is hard to read.**
>
> Thank you for pointing this out. In the revision we have reorganized Table 2 to present a systematic, cross-dataset evaluation.
>
> Concretely:
>
> * We now run additional experiments on SVHN, MNIST, and CIFAR-100 across multiple heterogeneity levels ($\alpha$).
> * We split the original “all-in-one” table into separate tables:
>   (i) low $\alpha$ across datasets,
>   (ii) high $\alpha$ across datasets,
>   (iii) across architectures, and
>   (iv) across labeling regimes (low vs. high budget, newly added).
>   This makes the behavior of AHFAL clearer and easier to interpret.
>
> Across all the additional results, we observe the same pattern as on CIFAR-10:
>
> * Under high heterogeneity (small $\alpha$), AHFAL outperforms prior centralized and federated active learning methods.
> * Under low heterogeneity (larger $\alpha$), all methods become closer in performance, and AHFAL remains comparable to or better than the strongest baselines.
>
> We hope this improved structure helps with the clarity of the results.

---

> ### Author Response · Authors · 2025-11-21
>
> **Q8. Instead of using entropy for sampling in Equation (3), have you considered alternative sampling strategies such as margin sampling?**
>
> We thank the reviewer for this suggestion. In this paper we instantiate AHFAL with entropy for two reasons: (i) it is the most commonly used and well-understood uncertainty measure in federated active learning, which makes comparisons to prior work clearer, and (ii) it simplifies the exposition of our bias–variance model in Section 5.
>
> Conceptually, however, AHFAL is agnostic to the specific uncertainty heuristic: Eq. (3) only requires a scalar score from a local and a global model. One can replace $H_L(x)$ and $H_G(x)$ by any acquisition score, including margin sampling [3] (or BALD, variation ratios, etc.), and the rest of the algorithm (class-variance-based routing between local vs. hybrid, and class-wise budget allocation) remains unchanged.
>
> We have clarified this modularity in the revision (Section 6.1): follow up work can explore the optimal scoring and sampling strategies across heterogeneity regimes.
>
> **Q9. How many datapoints do you have for Figure 3? The line looks straight, is it just 4 datapoints?**
>
> We thank the reviewer for this question and the opportunity to clarify Figure 3.
>
> Figure 3 is not based on four datapoints. For each heterogeneity level alpha and each active learning strategy, we compute pairs of (i) the standard deviation describing how each class is distributed across clients and (ii) the corresponding test accuracy achieved by that strategy.
>
> Since we have 10 classes (with typically different inter-client variances), 6 heterogeneity levels, 7 strategies, and 3 independent trials, this results in a dense scatter plot with thousands of data points overall.
>
> To analyze how decentralized and centralized strategies differ with respect to the correlation between class-distribution variance and accuracy, we group strategies by type (centralized vs. decentralized) and overlay a linear trend line for each group to summarize the overall relationship. This has been clarified in the revision, in the figure caption as well as the paper text (Section 4.2).

---

> > ### Comment · Reviewer_vAKh · 2025-11-27
> >
> > Thank you for the clarifications and the additional data. I recommend including the additional data (budgets) in the appendix, and clearly specifying in the main text which datasets were used for hyperparameter tuning and which ones relied on transferred hyperparameters.

---

> > > ### Author Response · Authors · 2025-12-03
> > >
> > > Thank you for the follow-up and for the concrete suggestions. We are glad that the reviewer's comments and concerns have been resolved.
> > >
> > > In the revised manuscript, we have addressed both these final points: (i) we have moved the additional labeling-budget experiments to the Appendix for completeness, and (ii) we now explicitly state in the main text how the threshold τ is selected, including which dataset(s)/setting(s) are used for tuning versus which evaluations use transferred hyperparameters. We hope these changes fully resolve the concerns about budget breadth and hyperparameter generalization, and contribute positively to the final evaluation of this work.

---

### Official Review · Reviewer_XryK · 2025-10-30

**Soundness:** 3
**Presentation:** 3
**Contribution:** 2
**Rating:** 4
**Confidence:** 3

**Summary:**

This paper focuses on active learning in federated settings with non-iid data distribution. The authors conducted a systematic analysis and reveals three important findings. Based on these, the authors propose a novel federated active learning method and valiate the proposed method on 4 datasets.

**Strengths:**

1. Generally, this paper is easy to follow.

2. The authors focuses on an important problem in federated learning, heterogeneous settings.

**Weaknesses:**

1. While Section 5 offers some theoretical insights into entropy estimation under heterogeneity, the analysis feels a bit surface-level and doesn’t clearly explain the reasoning behind AHFAL’s specific design choices. The link between the MSE analysis and the practical threshold $\tau$ isn’t very clear, which makes the theory section feel somewhat detached from the main algorithm.

2. The authors are supposed to compare the proposed method with more sota methods in the field of federated active learning, such as [1]. In addition, it misses out on more recent adaptive active learning and meta-learning–based selection strategies that might handle heterogeneity in different ways.

[1] Yingpeng Tang, Chao Ren, Xiaoli Tang, Sheng-Jun Huang, Lizhen Cui & Han Yu, "Efficient Heterogeneity-Aware Federated Active Data Selection," in Proceedings of the 42nd International Conference on Machine Learning (ICML'25), 2025.

3. The method involves computing and sending class distributions, generating pseudo-labels for all unlabeled samples, and possibly querying both local and global models. The paper calls the overhead “lightweight,” but there’s no actual measurement to back that up. It’s also unclear how well this would scale to thousands of clients or really large unlabeled datasets.

**Questions:**

1. Can you provide more rigorous justification for the fixed $\lambda = 1/2$ choice in Eq. (3)?

2. You mention plans to extend the approach to other domains. What do you see as the main challenges in adapting AHFAL to structured output tasks like object detection or segmentation, where the class boundaries aren’t as clear-cut?

---

> ### Author Response · Authors · 2025-11-21
>
> We thank all reviewers for their careful reading and constructive feedback. We are encouraged that the central insight of the paper, that class-wise data heterogeneity is a fundamental driver of which federated active learning strategy works best and that the strategy should adapt at the class level, is viewed as novel and well supported by our analysis and experiments (R fNx8, R vAKh). We also appreciate the positive comments that the paper is well written, easy to follow, and clearly structured, with figures that help convey the main findings (R BM9g, R XryK, R vAKh).
>
> Reviewers further noted that AHFAL is simple and practical, relying only on low-dimensional class histograms that fit naturally into standard FL workflows while admitting privacy-preserving mechanisms (R fNx8, R vAKh). The method is seen as effective and fairly general, with strong performance improvements over both centralized and federated baselines across datasets and architectures, and a code release that supports reproducibility and follow-up work (R fNx8, R BM9g, R vAKh). We now respond point by point to specific questions.
>
> **Q1. While Section 5 offers some theoretical insights into entropy estimation under heterogeneity, the analysis feels a bit surface-level and doesn’t clearly explain the reasoning behind AHFAL’s specific design choices. The link between the MSE analysis and the practical threshold  isn’t very clear, which makes the theory section feel somewhat detached from the main algorithm.**
>
> We thank the reviewer for this comment. The goal of Section 5 is not to provide full generalization bounds, but to give a simple bias–variance model that explains why
> (i) mixing local and global entropies can help, and
> (ii) this mixing should depend on class-wise heterogeneity.
>
> For class $c$ on client $i$, we model local and global entropy estimators as $\hat H^L\_c(x) = H^\*(x) + b^L\_{i,c} + \varepsilon^L\_{i,c}(x)$ and $\hat H^G\_c(x) = H^\*(x) + \beta\_c + \varepsilon^G\_c(x)$, where $b^L\_{i,c}$ is a client-specific bias and $\beta\_c$ is a class-wise global bias induced by cross-client imbalance.
>
> For any convex combination $\hat H^{(\lambda)}\_c(x) = \lambda \hat H^L\_c(x) + (1-\lambda)\hat H^G\_c(x)$, we derive the MSE-optimal mixing weight $\lambda^\* = \frac{V\_G - \rho + \beta\_c(\beta\_c - b^L\_{i,c})}{V\_L + V\_G - 2\rho + (b^L\_{i,c} - \beta\_c)^2}$ and show that a hybrid estimator $(0 < \lambda^\* < 1)$ is preferable to purely local estimation $(\lambda^\* = 1)$ whenever $\beta\_c^2 < 3V\_L - V\_G - 2\rho$.
>
> We refer to this condition as (★).
>
> AHFAL is a direct discretization of this condition:
> * We cannot observe $\beta_c$ directly, but our analysis (Section 6 and Appendix) shows that the global class bias $|\beta_c|$ grows with cross-client dispersion of class $c$. We therefore use the empirical class variance $\sigma_c$ as a monotone proxy for $\beta_c^2$.
> * Condition (★) partitions classes into two regimes: one where hybrid entropy has lower MSE and one where purely local entropy is better. AHFAL implements exactly this partition via a threshold on $\sigma_c$: classes with $\sigma_c < \tau$ are treated as satisfying (★) and use a hybrid estimator (we fix $\lambda \approx 1/2$), while classes with $\sigma_c \ge \tau$ are treated as violating (★) and use purely local entropy $(\lambda = 1)$.
> * The exact boundary in (★) depends on unobserved quantities $V_L, V_G, \rho$, so we use a single scalar $\tau$ as a calibrated surrogate. Our ablations show that performance is stable over a wide range of $\tau$, indicating that AHFAL mainly relies on the relative separation between low- and high-variance classes predicted by (★), rather than on fine-tuning the threshold.
>
>
>
> Consistent with this, we have added an additional paragraph at the end of Section 5 in the paper to further emphasize this relevance.

---

> ### Author Response · Authors · 2025-11-21
>
> **Q2. The method should be compared against more state-of-the-art FAL algorithms, such as Tang et al. (ICML 2025), and should also consider more recent adaptive / meta-learning–based active learning strategies that might handle heterogeneity differently.**
>
> We thank the reviewer for this suggestion and agree that broadening the set of baselines strengthens the paper. For Tang et al. (“Efficient Heterogeneity-Aware Federated Active Data Selection”, ICML 2025), we could not find publicly available code to reproduce their method faithfully in our setting. However, we fully agree with the spirit of the comment and have expanded our experimental comparison with two recent and highly relevant methods:
>
> * **IFAL** (Inconsistency-based Federated Active Learning, IJCAI 2025): a state-of-the-art FAL method that directly targets label heterogeneity via inconsistency detection.
> * **ACAL** (Adaptive Curriculum Active Learning, MICCAI 2024): a recent adaptive active learning strategy that explicitly handles heterogeneity and can be naturally instantiated in a federated setup.
>
> With these additions, our empirical evaluation now includes **10 baselines** spanning centralized AL, federated AL, and adaptive/meta-AL strategies. On CIFAR-10, AHFAL outperforms both IFAL and ACAL under the heterogeneous regimes considered. For example:
>
> * **CIFAR-10, ($\alpha = 0.1$)**
>
>   * AHFAL: (66.15 $\pm$ 0.94)
>   * ACAL: (56.87 $\pm$ 3.11)
>   * IFAL: (58.51 $\pm$ 3.35)
>
> * **CIFAR-10, ($\alpha = 0.3$)**
>
>   * AHFAL: (77.26 $\pm$ 0.45)
>   * ACAL: (74.54 $\pm$ 0.78)
>   * IFAL: (73.26 $\pm$ 0.36)
> If the reviewers deem these baselines useful, we can incorporate them in the revision.
> Separately, we also note that Tang et al. primarily compare against LoGo as the only federated active learning method, which is already one of our original eight baselines. Our expanded set therefore provides a more comprehensive and up-to-date benchmark for AHFAL, covering both dedicated FAL methods and adaptive selection strategies.
>
>
> **EDIT by the authors:** During the discussion phase, we extended this comparison with additional experiments for IFAL and ACAL across multiple datasets and heterogeneity levels. The detailed additional results are:
>
> * **CIFAR-10, ($\alpha = 0.5$)**
>   * AHFAL: (79.10 $\pm$ 0.47)
>   * ACAL: (76.44 $\pm$ 0.88)
>   * IFAL: (74.88 $\pm$ 0.14)
>
> * **CIFAR-10, ($\alpha = 1.0$)**
>   * AHFAL: (79.82 $\pm$ 0.39)
>   * ACAL: (78.49 $\pm$ 0.32)
>   * IFAL: (76.27 $\pm$ 0.63)
>
> * **SVHN, ($\alpha = 0.1$)**
>   * AHFAL: (85.61 $\pm$ 2.07)
>   * ACAL: (75.95 $\pm$ 10.15)
>   * IFAL: (75.33 $\pm$ 4.16)
>
> * **SVHN, ($\alpha = 1.0$)**
>   * AHFAL: (94.35 $\pm$ 0.17)
>   * ACAL: (94.22 $\pm$ 0.25)
>   * IFAL: (92.67 $\pm$ 0.10)
>
> Across all these settings, AHFAL consistently outperforms both IFAL and ACAL, supporting the robustness of our conclusions.

---

> ### Author Response · Authors · 2025-11-21
>
> **Q3. The method requires computing and transmitting class distributions, generating pseudo-labels for all unlabeled samples, and sometimes querying both local and global models. The paper calls this “lightweight” without measurements, and it is unclear how this scales to many clients or very large unlabeled pools.**
>
> We thank the reviewer for raising this point. We have now quantified both the communication and computational overhead of AHFAL and compared it to standard FAL baselines. We have incorporated these insights into the revised manuscript in Subsection 7.4.
>
> **Communication overhead**
>
> The additional communication introduced by AHFAL comes only from transmitting class histograms, which is negligible compared to standard model updates in FL.
>
> In our CIFAR-10 setup (10 classes, 10 clients):
>
>   * Per client, per cycle: a 10-dimensional float vector ($\mathbf{p}_i$)
>     ($\Rightarrow$) (10 $\times$ 4) bytes (= 40) bytes.
>   * Across 10 clients: (10 $\times$ 40 = 400) bytes per cycle.
>   * Model transmission per cycle: roughly (311 KB) per client
>     ($\Rightarrow$) ( $\approx$ 3.12 MB) across 10 clients.
>   * Overhead ratio: $\frac{400\ \text{bytes}}{3.12 \times 10^6\ \text{bytes}} \approx 0.013\%$.
>
>
> **Scaling to larger systems:**
>   With (C) classes and (N) clients, the histogram cost scales as (O(NC)).
>   Even with (N = 1000) clients and (C = 100) classes, each client sends
>   (100 $\times$ 4 = 400) bytes, for a total of $(\approx$ 400 KB ) per cycle.
>   Compared to typical FL model updates (tens–hundreds of MB for realistic models), this remains well below **0.1%** of the model communication cost.
>
> Thus, the extra communication required by AHFAL is several orders of magnitude smaller than the standard FL communication and remains negligible even at large scale.
>
> **Computational overhead**
>
> On the computation side, AHFAL is comparable to existing FAL methods and significantly cheaper than the heaviest baselines.
>
> * **Uncertainty scoring:**
>   AHFAL requires:
>
>   * one forward pass through the local model for all unlabeled samples (to get pseudo-labels and (H^L)), and
>   * a forward pass through the global model only for samples whose pseudo-labels belong to low-variance classes.
>     In the worst case, this corresponds to at most two forward passes per sample, which is similar to KAFAL, where KL divergences between local and global predictions are computed.
>
> * **Histogram and routing logic:**
>   Class-histogram aggregation and the ($\sigma_c$)-based partitioning contribute only (O(NC)) operations per round, which amounts to **< 0.1 ms** per cycle in our implementation; this is negligible relative to local training and inference over the unlabeled pool.
>
> * **Wall-clock measurements** (CIFAR-10, ($\alpha = 0.1$), averaged over 3 trials):
>
>   * Random: ~5h 24m
>   * Entropy: ~5h 29m
>   * Core-Set: ~5h 30m
>   * LoGo: ~5h 28m
>   * KAFAL: ~5h 27m
>   * FEAL: ~5h 24m
>   * Noise Stability: ~6h 24m
>   * **AHFAL:** ~5h 34m
>   * BADGE: ~7h 25m
>
> AHFAL’s runtime is within a few minutes of KAFAL, Entropy, and Core-Set, and substantially faster than BADGE, whose gradient-based, high-dimensional clustering incurs much higher computational cost. This empirically supports our claim that AHFAL’s extra logic is lightweight relative to the dominant forward/backward passes already present in FAL pipelines.
>
> We have added these communication and computation measurements, as well as the (O(NC)) scaling analysis, to the revised version (Section 7.4) to make the “lightweight overhead” claim precise and transparent for large-scale settings.
>
>
> **Q4. Can you provide more rigorous justification for the fixed choice of lambda in Eq. (3)?**
>
> The fixed coefficient (1/2) in Eq. (3) is directly motivated by our bias–variance model. We model local and global entropy estimators for class $c$ on client $i$ as $\hat H^L\_c(x) = H^\*(x) + b^L\_{i,c} + \varepsilon^L\_{i,c}(x)$ and $\hat H^G\_c(x) = H^\*(x) + \beta\_c + \varepsilon^G\_c(x)$, and consider their convex combination $\hat H^{(\lambda)}\_c(x) = \lambda \hat H^L\_c(x) + (1-\lambda)\hat H^G\_c(x)$.
>
> From the resulting MSE expression, the optimal mixing weight $\lambda^\*$ lies strictly between 0 and 1 whenever the global bias $\beta\_c$ is not too large, and in the *symmetric* regime where local and global estimators have comparable bias and variance (precisely the low-variance classes where we use the hybrid estimator), $\lambda^\*$ concentrates near $1/2$.
>
> Moreover, the MSE is quadratic in $\lambda$, so the excess error away from $\lambda^\*$ scales as $(\lambda - \lambda^\*)^2$; choosing the symmetric point $\lambda = 1/2$ therefore yields a simple, closed-form choice that is near-optimal throughout the regime where hybridization is theoretically preferable, while avoiding the need to estimate class- and client-specific $\lambda^\*$. This provides a rigorous justification for the fixed coefficient used in Eq. (3).
>
>
> We have revised the paper (Section 5) to make this justification clearer.

---

> ### Author Response · Authors · 2025-11-21
>
> **Q5. What do you see as the main challenges in adapting AHFAL to structured output tasks like object detection or segmentation, where the class boundaries aren’t as clear-cut?**
>
> Thank you for this thoughtful question. We fully agree that extending AHFAL to structured outputs is an interesting direction. Conceptually, the framework carries over, but there are a few technical challenges we would need to address:
>
> 1. **Defining heterogeneity.**
>    In classification we use per-class label histograms; for dense outputs we must decide whether to measure imbalance over objects, pixels, regions, or learned prototypes in feature space.
> 2. **Acquisition granularity.**
>    AHFAL currently selects whole samples; structured tasks need region-/patch-level acquisition and corresponding “counts” so that local vs. hybrid decisions are well-defined at that finer granularity.
> 3. **Scalable statistics.**
>    The effective label space (classes × scales × context) is much larger, so we need compressed, privacy-preserving summaries (e.g., clustered prototypes) to track heterogeneity without excessive communication.
>
> The underlying principle remains the same: rare, client-specific patterns should rely more on local models, while common, well-shared patterns can benefit from hybrid local–global estimates, but instantiating this for structured outputs is an important direction for future work, and we appreciate the reviewer highlighting it.

---

### Official Review · Reviewer_BM9g · 2025-10-31

**Soundness:** 3
**Presentation:** 3
**Contribution:** 3
**Rating:** 6
**Confidence:** 4

**Summary:**

This paper proposes a new algorithm, Adaptive Hybrid Federated Active Learning (AHFAL), to dynamically integrate centralized and decentralized paradigms for FAL. Based on their three key findings, they develop a new framework by grouping the clients into two disjoint sets and applying different scoring strategy. AHFAL showed good performance across diverse tasks.

**Strengths:**

- Their method is built on top of reasonable and good findings.
- The performance improvement over various baselines seems also quite good.
- The overall paper is well-written and has good structure.

**Weaknesses:**

- I'm quite confused with the definition of "centralized" and "decentralized" algorithms. There might be better words for them I believe.
- Could you elaborate Figure 4 experiments more? I have no idea what this figure means exactly and how the finding 3 comes out.
- I'm personally fine with communicating the client distribution to the server (privacy issue may be little), but it would be good to note that which methods use client distribution like this. Afaik, some decentralized strategy like LoGo does not share client distribution using the server.
- Do you have any other results to show compatibility with other FL algorithms than FedAvg?

**Questions:**

See above Weakness section.

---

> ### Author Response · Authors · 2025-11-21
>
> We thank all reviewers for their careful reading and constructive feedback. We are encouraged that the central insight of the paper, that class-wise data heterogeneity is a fundamental driver of which federated active learning strategy works best and that the strategy should adapt at the class level, is viewed as novel and well supported by our analysis and experiments (R fNx8, R vAKh). We also appreciate the positive comments that the paper is well written, easy to follow, and clearly structured, with figures that help convey the main findings (R BM9g, R XryK, R vAKh).
> Reviewers further noted that AHFAL is simple and practical, relying only on low-dimensional class histograms that fit naturally into standard FL workflows while admitting privacy-preserving mechanisms (R fNx8, R vAKh). The method is seen as effective and fairly general, with strong performance improvements over both centralized and federated baselines across datasets and architectures, and a code release that supports reproducibility and follow-up work (R fNx8, R BM9g, R vAKh). We now respond point by point to specific questions.
>
> **Q1. The terms “centralized” and “decentralized” are confusing in this context, and there may be more other, more appropriate wording.**
>
> We appreciate this comment and agree that our terminology can be clearer. In the current draft, we use “centralized” for methods where each client runs a standard active learning strategy in isolation, without using any cross-client information beyond the usual federated model updates. We use “decentralized” for methods whose acquisition decisions explicitly depend on aggregated cross-client information (for example, a shared global model or global class statistics), hence they are aware of the federated structure and heterogeneity.
>
> We chose this wording to reflect the distinction between purely local decision-making and cross-client coordination, but we agree that “centralized/decentralized” is overloaded in the FL literature and may suggest raw data centralization, which we do not perform.
>
> In the revision, we are happy to adopt alternative labels, for example:
> “Local-only” vs. “global-aware” active learning, or
> “Client-isolated” vs. “cross-client” active learning.
> Based on reviewer feedback, we will update the terminology consistently in the text and figures.
>
> **Q2. Elaborating on Figure 4 experiments and emphasizing the connection to Finding 3.**
>
> Figure 4 compares three quantities over active learning cycles in the same federated setting: the mean accuracy across all standard baselines (“Average Baseline”), the best-performing baseline among them (“Best Baseline”), and our oracle variant (“Global Class Distribution Labeling”), where clients are given the true global class histogram and use it to bias their label budget toward classes that are underrepresented locally relative to the global distribution (while still ranking samples within each class by the same uncertainty score as the baselines).
>
> Finding 3 comes directly from the consistent vertical gap between the orange curve and the yellow “Best Baseline” curve across cycles: for each cycle, the oracle variant achieves higher accuracy, and this improvement is roughly 2–3 percentage points on average. Because the acquisition heuristic and model are unchanged and only the class-wise allocation is modified using global class-distribution information, this figure isolates and quantifies the benefit of having access to global class statistics, which is exactly what Finding 3 claims.

---

> ### Author Response · Authors · 2025-11-21
>
> **Q3. Communicating the client distribution to the server sounds reasonable (privacy issue may be little), but it would be good to note which methods use client distribution like this.**
>
> We thank the reviewer for pointing this out and will clarify this aspect. To the best of our knowledge, none of the compared prior FAL methods (including LoGo, KAFAL, FEAL) explicitly communicate per-client class histograms to the server in the way AHFAL does.
>
> This lightweight sharing of local class distributions is in fact a key design choice and source of novelty in our method, as it enables the class-wise heterogeneity analysis and routing rule. As part of this revision, we are now also reporting the additional computation and communication overhead of sending these low-dimensional histograms and show that it is negligible compared to standard FL traffic, while yielding both the theoretical and empirical gains we highlight.
>
> **Q4. Do we have any other results to show compatibility with other FL algorithms than FedAvg?**
>
> In our current experiments we use FedAvg as the default FL optimizer, but AHFAL is designed to be orthogonal to the aggregation rule. The algorithm only assumes that at each round (i) clients train a local model on their labeled data and (ii) a single global model snapshot is produced and broadcast. All of our active learning logic (class-variance estimation, local vs. hybrid entropy, and class-wise budget allocation) operates on top of whatever global model the FL method provides, without modifying the aggregation itself.
>
> This means AHFAL can be plugged into alternatives such as FedProx, FedNova, etc., by simply replacing the aggregation step while keeping the acquisition procedure unchanged. We already demonstrate this modularity by combining AHFAL with KCFU (Table 8, Appendix), where the update mechanism is swapped without changing our selection rule. We are happy to include additional experiments in a further revision if the reviewers recommend it.

---

> > ### Comment · Reviewer_BM9g · 2025-11-25
> >
> > Thanks for the responses. All my concerns are well-addressed.
> > I'll keep my score accordingly.

---

> > > ### Author Response · Authors · 2025-12-03
> > >
> > > Thank you for the positive feedback. We’re glad the revisions addressed all of your concerns, and we hope these clarifications and additions contribute positively to the final evaluation of the work.

---

### Official Review · Reviewer_fNx8 · 2025-11-01

**Soundness:** 3
**Presentation:** 3
**Contribution:** 2
**Rating:** 2
**Confidence:** 4

**Summary:**

This paper tackles the problem of Federated Active Learning under data heterogeneity. The authors observe that: the optimal active learning strategy depends on the degree of data heterogeneity. Specifically, decentralized methods excel in low-heterogeneity settings, while simple centralized methods surprisingly outperform them when heterogeneity is high. The paper proposes AHFAL: classes that are "high-variance" (i.e., concentrated on only a few clients) are better served by local-only sampling, while "low-variance" classes (i.e., spread evenly across clients) benefit from a hybrid local-global strategy. The authors provide empirical analysis to motivate their approach and show that AHFAL outperforms several centralized and decentralized baselines, particularly in high-heterogeneity regimes.

**Strengths:**

1. The central finding that the optimal FAL strategy is not static but should be adaptive at the class level based on distribution variance, is novel and intuitive. This moves beyond the typical FAL approach of finding a single "best" way to combine local and global information.

2. The proposed method is practical. The "cost" is sharing local class histograms, which are low-dimensional vectors. The authors discuss the privacy implications and potential mitigations.

3. The paper compares against a good set of baselines, including both centralized (Entropy, BADGE, Core-Set) and SOTA decentralized (LoGo, KAFAL, FEAL) methods. The codebase is also provided for reproducibility.

**Weaknesses:**

1. My primary concern is that the algorithmic novelty may not meet the threshold for ICLR. The main algorithmic step appears to be a class-wise recombination of existing paradigms rather than a fundamentally new mechanism. Specifically, the method adopts existing strategies on a per-class basis. Furthermore, the concept of exploiting both local and global models to mitigate heterogeneity has been explored in prior work [1, 2].

2. The core analysis section motivating the method runs only on CIFAR-10. This limited empirical grounding weakens the causal link between the initial findings and the subsequent generalization of the proposed framework.

3. The experimental validation is not sufficiently robust. While CIFAR-10 is analyzed thoroughly, other datasets (CIFAR-100, SVHN, MNIST) are introduced sporadically in figures and tables without clear explanation or systematic analysis. This lack of consistent evaluation across all benchmarks diminishes the robustness of the paper's empirical claims.

[1] Cao, Yu-Tong, et al. "Knowledge-aware federated active learning with non-iid data." Proceedings of the IEEE/CVF International Conference on Computer Vision. 2023.

[2] Deng, Zhipeng, et al. "Fedal: An federated active learning framework for efficient labeling in skin lesion analysis." 2022 IEEE International Conference on Systems, Man, and Cybernetics (SMC). IEEE, 2022.

**Questions:**

Please see weakness.

---

> ### Author Response · Authors · 2025-11-21
>
> We thank all reviewers for their careful reading and constructive feedback. We are encouraged that the central insight of the paper, that class-wise data heterogeneity is a fundamental driver of which federated active learning strategy works best and that the strategy should adapt at the class level, is viewed as novel and well supported by our analysis and experiments (R fNx8, R vAKh). We also appreciate the positive comments that the paper is well written, easy to follow, and clearly structured, with figures that help convey the main findings (R BM9g, R XryK, R vAKh).
>
> Reviewers further noted that AHFAL is simple and practical, relying only on low-dimensional class histograms that fit naturally into standard FL workflows while admitting privacy-preserving mechanisms (R fNx8, R vAKh). The method is seen as effective and fairly general, with strong performance improvements over both centralized and federated baselines across datasets and architectures, and a code release that supports reproducibility and follow-up work (R fNx8, R BM9g, R vAKh). We now respond point by point to specific questions.
>
> **Q1. The main algorithmic step appears to be a class-wise recombination of existing paradigms rather than a fundamentally new mechanism, and similar ideas of combining local and global models to handle heterogeneity have been explored in prior work [Cao et al. 2023; Deng et al. 2022], so the algorithmic novelty may not meet the ICLR threshold.**
>
>
> We respectfully disagree that the contribution of this work is limited to a minor recombination of existing FAL mechanisms. Simply "using both local and global models" is indeed common to virtually all federated active learning methods; by definition, every FAL algorithm relies on local updates plus a global model and must cope with heterogeneity. We do not claim novelty at that level.
>
> Our contribution is threefold, with the algorithm intentionally kept simple:
>
> * **Class-wise heterogeneity as the key driver.**
> We show that which strategy wins (local-only vs federated-aware) flips as a function of class-wise cross-client variance, not just client-level heterogeneity. This leads to a new, explicit per-class routing rule: high-variance classes use purely local scores, low-variance classes use a hybrid local + global score. Prior methods such as KAFAL and FedAL use a single strategy that does not adapt by class.
>
> * **Theory that directly specifies the mechanism.**
> Our bias–variance analysis for local and global entropy estimators yields a condition under which a hybrid estimator is preferable to a local one, in terms of a class-specific global bias. We connect this bias to cross-client class variance and discretize it into the thresholding rule and fixed mixing in AHFAL. The algorithm is thus a direct instantiation of this analysis, not an ad hoc blend of heuristics.
>
> * **Contribution aligned with “meta active learning.”**
> Recent work in active learning emphasizes understanding, selecting, and combining existing strategies for specific regimes, rather than defining new uncertainty scores. In this spirit, our main contribution is (i) diagnosing how FAL behavior depends on class-wise heterogeneity and (ii) providing a simple, plug-in class-wise adaptive mechanism that consistently improves over centralized and federated baselines across datasets and architectures.
>
> In summary, the novelty of the paper does not rest solely on introducing a complex new algorithm. It lies in identifying class-wise heterogeneity as the key driver of FAL behavior, providing a theoretical link between this quantity and the benefit of hybrid vs local entropy, and translating that into a deliberately simple, practical mechanism that is easy to adopt in real federated systems, consistent with other accepted work in active learning in top venues, but novel in the space of federated active learning.

---

> ### Author Response · Authors · 2025-11-21
>
> **Q2. The core empirical analysis that motivates AHFAL (e.g., the centralized–decentralized trade-off) is shown only on CIFAR-10. Can the same analysis be demonstrated on additional datasets to support its generality?**
>
> Thank you for this helpful suggestion. We focused the main-text analysis on CIFAR-10 for readability: mixing multiple datasets on a single plot requires different accuracy scales and can obscure the class- and heterogeneity-dependent effects we want to highlight.
> That said, we have repeated the core “Finding 1” analysis (centralized vs. decentralized trade-off across heterogeneity levels) on SVHN and MNIST. In both cases we observe the same qualitative behavior: centralized methods dominate under strong heterogeneity, while decentralized methods catch up and can slightly surpass centralized ones as heterogeneity decreases.
>
> **SVHN (Finding 1 replicated)**
>
> | ($\alpha$) | Centralized | Decentralized |
> | -------- | ----------- | ------------- |
> | 0.1      | 79.93       | 69.56         |
> | 0.5      | 91.08       | 91.73         |
> | 1.0      | 93.18       | 93.89         |
>
> **MNIST (Finding 1 replicated)**
>
> | ($\alpha$) | Centralized | Decentralized |
> | -------- | ----------- | ------------- |
> | 0.1      | 82.13       | 79.97         |
> | 0.5      | 99.05       | 99.10         |
> | 1.0      | 98.41       | 98.44         |
>
> These results confirm that the heterogeneity-driven trade-off between centralized and decentralized sampling is **not specific to CIFAR-10**, but holds across additional datasets with very different difficulty and class structure. If the reviewers find it useful, we will include these multi-dataset analyses (and extend them to the other findings) in the appendix of the final revision.
>
> **EDIT by the authors:** During the discussion phase, we followed up on this plan and extended the core empirical analysis to both key findings: (1) the heterogeneity-driven trade-off between centralized and decentralized sampling, and (2) the class-wise variance analysis explaining the crossover behavior.
>
> We conducted these analyses on two additional datasets, SVHN and MNIST, and incorporated the complete multi-dataset results into Appendix M of the revised paper.
>
> These additional results are consistent with our observations on CIFAR-10, further strengthening the empirical support for our insights.

---

> ### Author Response · Authors · 2025-11-21
>
> **Q3. While CIFAR-10 is analyzed thoroughly, other datasets (CIFAR-100, SVHN, MNIST) are introduced sporadically in figures and tables without clear explanation or systematic analysis.**
>
> Thank you for pointing this out. In the revision we have strengthened and reorganized the results section to present a systematic, cross-dataset evaluation rather than sporadic examples.
>
> Concretely:
>
> * We now run additional experiments on SVHN, MNIST, and CIFAR-100 across multiple heterogeneity levels ($\alpha$).
> * We split the original “all-in-one” table into separate tables:
>   (i) low $\alpha$ across datasets,
>   (ii) high $\alpha$ across datasets,
>   (iii) across architectures, and
>   (iv) across labeling regimes (low vs. high budget, newly added).
>   This makes the behavior of AHFAL clearer and easier to interpret.
>
> Across all the additional results, we observe the *same pattern* as on CIFAR-10:
>
> * Under high heterogeneity (small $\alpha$), AHFAL outperforms prior centralized and federated active learning methods.
> * Under low heterogeneity (larger ($\alpha$)), all methods become closer in performance, and AHFAL remains comparable to or better than the strongest baselines.
>
> Below we summarize the new cross-dataset results (mean accuracy, %) that are included in the revised manuscript (Section 7.2 and Appendix F).
>
>
> ### **SVHN: Accuracy vs. heterogeneity**
>
> | Method   | ($\alpha = 0.1$) | ($\alpha = 0.5$) | ($\alpha = 1.0$) |
> | -------- | -------------- | -------------- | -------------- |
> | AHFAL    | 85.61      | 91.80          | 94.35      |
> | Entropy  | 85.31          | 92.15      | 93.85          |
> | Core-Set | 82.43          | 91.08          | 93.13          |
> | Noise    | 81.71          | 90.50          | 92.84          |
> | LoGo     | 80.84          | 91.59          | 93.56          |
> | BADGE    | 77.50          | 90.57          | 92.30          |
> | FEAL     | 69.98          | 92.08          | 94.21      |
> | Random   | 65.27          | 90.89          | 92.68          |
> | KAFAL    | 62.32          | 91.51          | 93.90          |
>
>
> ### **MNIST: Accuracy vs. heterogeneity**
>
> | Method   | ($\alpha = 0.1$) | ($\alpha = 0.5$) | ($\alpha = 1.0$) |
> | -------- | -------------- | -------------- | -------------- |
> | AHFAL    | 92.83      | 99.16      | 98.54          |
> | Entropy  | 87.63          | 99.20      | 98.63      |
> | Core-Set | 89.59          | 99.18          | 98.61          |
> | Noise    | 80.53          | 99.02          | 98.53          |
> | LoGo     | 90.52          | 99.21      | 98.41          |
> | BADGE    | 70.78          | 98.81          | 97.90          |
> | FEAL     | 72.72          | 99.06          | 98.55      |
> | Random   | 75.79          | 98.88          | 97.96          |
> | KAFAL    | 76.68          | 99.02          | 98.37          |
>
>
> ### **CIFAR-100: Accuracy vs. heterogeneity**
>
> | Method   | ($\alpha = 0.5$) | ($\alpha = 1.0$) |
> | -------- | -------------- | -------------- |
> | AHFAL    | 43.10          | 44.03      |
> | Entropy  | 42.73          | 42.94          |
> | Core-Set | 43.10      | 43.98          |
> | Noise    | 42.45          | 42.98          |
> | LoGo     | 43.46          | 43.93          |
> | BADGE    | 41.00          | 41.71          |
> | FEAL     | 42.06          | 42.23          |
> | Random   | 42.89          | 43.32          |
> | KAFAL    | 43.33     | 43.47          |
>
> We hope this more structured presentation addresses the reviewer’s concern about systematic multi-dataset analysis. We are happy to incorporate any additional reviewer feedback to further improve on this.

---

### Author Response · Authors · 2025-11-21
**Summary of revisions**

We thank all reviewers for their time, careful reading, and constructive feedback. In response, we have revised the manuscript and uploaded an updated PDF. All additions and edits in both the main paper and appendix are highlighted in blue for ease of inspection. The main changes are:

* **Figure 3 clarification:** We now explicitly describe the number of data points and the procedure used to generate Figure 3 (class-wise variance vs. accuracy).
* **Justification of ($\lambda = 1/2$):** Section 5 has been expanded to provide a more rigorous justification for the fixed mixing coefficient in Eq. (3) using our bias–variance analysis.
* **Linking theory to AHFAL:** We clarify how the theoretical MSE result directly informs the class-partitioning and routing rules in AHFAL (Section 5).
* **Acquisition function generality:** Section 6 now explains that while we instantiate AHFAL with entropy-based scoring, the framework is agnostic to the specific uncertainty heuristic and can incorporate alternatives such as margin sampling.
* **Restructured and extended results tables:** The original Table 2 has been split into Tables 2, 3, and 4 to improve readability, and now includes additional results beyond CIFAR-10 across different heterogeneity levels.
* **Labeling budget analysis:** New experiments (Table 5) evaluate AHFAL under different labeling budgets (low- and high-budget regimes), showing that AHFAL remains superior across query schedules.
* **Overhead analysis:** A new Section 7.4 quantifies AHFAL’s communication and computation overhead and compares it to prior FAL methods.
* **Additional datasets and heterogeneity regimes:** Appendix F includes extended results for MNIST and SVHN across multiple heterogeneity levels, further validating the generality of our findings.

For each reviewer, we also provide point-by-point responses to all comments and questions, with explicit pointers to the corresponding changes in the revised manuscript.

---

### Author Response · Authors · 2025-12-03
**Final summary of revisions**

We thank all reviewers for their careful reading and constructive feedback. We are encouraged that the central insight of the paper, that class-wise data heterogeneity is a fundamental driver of which federated active learning strategy works best and that the strategy should adapt at the class level, is viewed as novel and well supported by our analysis and experiments (R fNx8, R vAKh). We also appreciate the positive comments that the paper is well written, easy to follow, and clearly structured, with figures that help convey the main findings (R BM9g, R XryK, R vAKh).

Reviewers further noted that AHFAL is simple and practical, relying only on low-dimensional class histograms that fit naturally into standard FL workflows while admitting privacy-preserving mechanisms (R fNx8, R vAKh). The method is seen as effective and fairly general, with strong performance improvements over both centralized and federated baselines across datasets and architectures, and a code release that supports reproducibility and follow-up work (R fNx8, R BM9g, R vAKh).

We have revised the manuscript and uploaded an updated PDF based on the feedback. All additions and edits in both the main paper and appendix are highlighted in blue for ease of inspection. The main changes are:
* **Figure 3 clarification:** We describe the procedure used to generate Figure 3 (class-wise variance vs. accuracy) in the response to Reviewer vAKh.
* **Justification of ($\lambda = 1/2$):** Section 5 has been expanded to provide a more rigorous justification for the fixed mixing coefficient in Eq. (3) using our bias–variance analysis.
* **Linking theory to AHFAL:** We clarify how the theoretical MSE result directly informs the class-partitioning and routing rules in AHFAL (Section 5).
* **Acquisition function generality:** Section 6 now explains that while we instantiate AHFAL with entropy-based scoring, the framework is agnostic to the specific uncertainty heuristic and can incorporate alternatives such as margin sampling.
* **Restructured and extended results tables:** The original Table 2 has been split into Tables 2, 3, and 4 to improve readability, and now includes additional results beyond CIFAR-10 across different heterogeneity levels.
* **Labeling budget analysis:** New experiments (Appendix G) evaluate AHFAL under different labeling budgets (low- and high-budget regimes), showing that AHFAL remains superior across query schedules. Earlier, this was table 5 in the main paper, but based on R vAKh’s feedback, we have added it to the appendix (discussed below).
* **Overhead analysis:** A new Section 7.4 quantifies AHFAL’s communication and computation overhead and compares it to prior FAL methods.
* **Additional datasets and heterogeneity regimes:** Appendix F includes extended results for MNIST and SVHN across multiple heterogeneity levels, further validating the generality of our findings.

In addition to the above changes, post the initial response, we have made additional updates:

* **Adding Labeling budget analysis to appendix:** Based on R vAKh’s concluding comment, we have added labelling budget analysis to the manuscript in appendix G to further emphasize the performance of AHFAL across varying constraints.
* **Clarifying hyperparamter turning approach:** Also based on R vAKh’s concluding comment, we have updated the main manuscript to clearly specify how the parameter $\tau$ is determined: specifically, determined for CIFAR-10 and directly used on other datasets.
* **Additional empirical analysis motivating the method:** To further address R fNx8’s comment on empirical analysis only being shown on CIFAR-10, we have gone ahead and carried out the empirical analysis of Section 4 specifically for Key Findings 1 and 2, on two additional datasets: SVHN and MNIST. These results are included in the Appendix M, and further support our findings from the empirical analysis, thereby providing additional grounding for AHFAL.
* **Additional baseline methods:** Based on feedback by R XryK, we ran additional experiments on two new baselines: IFAL and ACAL. Comparing with these recent methods across various conditions further emphasizes the superior performance of AHFAL. These results are currently in the updated response to R XryK and can be added to the final version of the paper.

For each reviewer, we also provided question-by-question responses.

We are glad that R BM9g and R vAKh have acknowledged that their concerns and comments have been adequately addressed by our response and associated changes, and we hope this acknowledgement contributes positively to the final evaluation of our work.
In addition, while R fNx8 and R XryK were unable to give feedback on our response, we believe the above mentioned updates (as well as the point-by-point responses made to the reviewers) address their comments and concerns, and we hope this also contributes positively to the final evaluation of our work.

---

### Meta-Review · Area_Chair_aLM4 · 2026-01-07

**Summary:**

The reviewers raised concerns about limited novelty, insufficient experimental evaluation, lack of computational complexity analysis, and limited scalability and compatibility. During the rebuttal, some minor issues were addressed (e.g., unclear figures and notations); however, the main concerns—specifically insufficient experiments—remain unresolved. Therefore, I recommend rejecting this paper.

**Reviewer Concerns:**

- Reviewer fNx8: limited novelty, insufficient experimental evaluation (e.g., only CIFAR10), limited robustness.
- Reviewer BM9g: unclear notations, unclear figures, limited compatibility of proposed method
- Reviewer XryK: insufficient theoretical justification, missing comparisons with SOTA methods, limited scalability.
- Reviewer vAKh: insufficient evaluation, narrow evaluation settings, unverified generalizability, lacks quantitative discussion of computational and communication overhead

**Reviewer Scores:**

- Reviewer fNx8: No
- Reviewer BM9g: Yes
- Reviewer XryK: No
- Reviewer vAKh: No

---

### Decision · Program_Chairs · 2026-01-26

Reject